# Improved GPCR ligands from nanobody tethering

Ross W. Cheloha [1], Fabian A. Fischer [1], Andrew W. Woodham[1], Eileen Daley[2], Naomi Suminski[1], Thomas J. Gardella[2 ✉] & Hidde L. Ploegh [1 ✉]

Antibodies conjugated to bioactive compounds allow targeted delivery of therapeutics to cell types of choice based on that antibody's specificity. Here we develop a new type of conjugate that consists of a nanobody and a peptidic ligand for a G protein-coupled receptor (GPCR), fused via their C-termini. We address activation of parathyroid hormone receptor-1 (PTHR1) and improve the signaling activity and specificity of otherwise poorly active N-terminal peptide fragments of PTH by conjugating them to nanobodies (VHHs) that recognize PTHR1. These C-to-C conjugates show biological activity superior to that of the parent fragment peptide in vitro. In an exploratory experiment in mice, a VHH-PTH peptide conjugate showed biological activity, whereas the corresponding free peptide did not. The lead conjugate also possesses selectivity for PTHR1 superior to that of PTH(1-34). This design approach, dubbed "conjugation of ligands and antibodies for membrane proteins" (CLAMP), can yield ligands with high potency and specificity.

[1] Boston Children's Hospital and Harvard Medical School, 1 Blackfan Circle, Boston, MA 02115, USA. [2] Massachusetts General Hospital and Harvard Medical School, 50 Blossom Street, Boston, MA 02114, USA. ✉email: gardella@helix.mgh.harvard.edu; hidde.ploegh@childrens.harvard.edu

Antibodies bind tightly and specifically to their targets, even in highly complex environments. This property of antibodies has been used to deliver bioactive compounds to sites of interest, both for diagnostic and therapeutic applications[1]. For example, conjugates between antibodies and cytotoxic drugs (antibody-drug conjugates or ADCs) can selectively kill cancer cells that display the antibody's target[2]. The success of ADCs often depends on the internalization of the conjugate through endocytosis, followed by release of the cytotoxic payload. Far fewer studies have made use of antibodies to deliver bioactive compounds with sites of action at the cell surface. The conjugation of a ligand for a surface receptor to an antibody that recognizes that same receptor should increase the effective concentration of the ligand and so increase its potency and specificity, provided appropriate spatial constraints are maintained. Ideally, this method could be used with an antibody that directly targets the receptor of interest to enable application without the need for genetic modification of the target cells or organism. The G protein-coupled receptor (GPCR) family of proteins is an attractive class of targets to pursue using this approach.

Molecules that target GPCRs represent more than 25% of all approved drugs[3]. Antibodies and the variable fragments of camelid heavy chain-only antibodies (VHHs or nanobodies) have found increasing use for modulating GPCR signaling[4,5]. GPCRs and their ligands display a considerable degree of degeneracy. Several natural ligands bind to more than a single GPCR and many GPCRs can bind more than one ligand[6,7]. The parathyroid hormone receptors constitute one such example: a bioactive N-terminal fragment of parathyroid hormone (PTH, residues 1–34), used under the name teriparatide to treat osteoporosis, potently activates both type-1 and type-2 PTH-receptors (PTHR1/PTHR2)[8]. PTHR1/2 are part of the class B of GPCRs which are naturally activated by large (>25 residue) peptides[9]. Despite intense industry interest, no small molecule agonists of B-family GPCRs with potencies comparable to the natural ligands have been described. To address PTHR signaling and selectivity, we prepared conjugates of fragments of PTH and VHHs. VHHs are appealing building blocks for these conjugates, as they are the smallest antibody fragments that retain the ability to bind antigens. They can be produced in high yield recombinantly[10]. The site of antigen recognition on VHHs is near the N-terminus[11,12] and the interaction of PTHR1 and PTHR2 with their ligands requires a free N-terminus on the latter[8,13,14]. Using a chemo-enzymatic approach we therefore made C-to-C-terminal fusions of PTH fragments and VHHs[15] to avoid any N-terminal obstructions. These chimeric molecules, dubbed: **c**onjugates of **l**igands and **a**ntibodies for **m**embrane **p**roteins or CLAMPs, target either wild-type or engineered receptor variants (Fig. 1). The optimized CLAMPs display biological activities in vitro and in vivo that are superior to those of the PTH fragments from which they were derived. When otherwise weakly active PTH fragments are incorporated into these conjugates, they can be made exquisitely selective for activation of only those receptors engaged by the VHH. This stands in marked contrast to the lack of selectivity shown by PTH(1-34)[8]. These findings

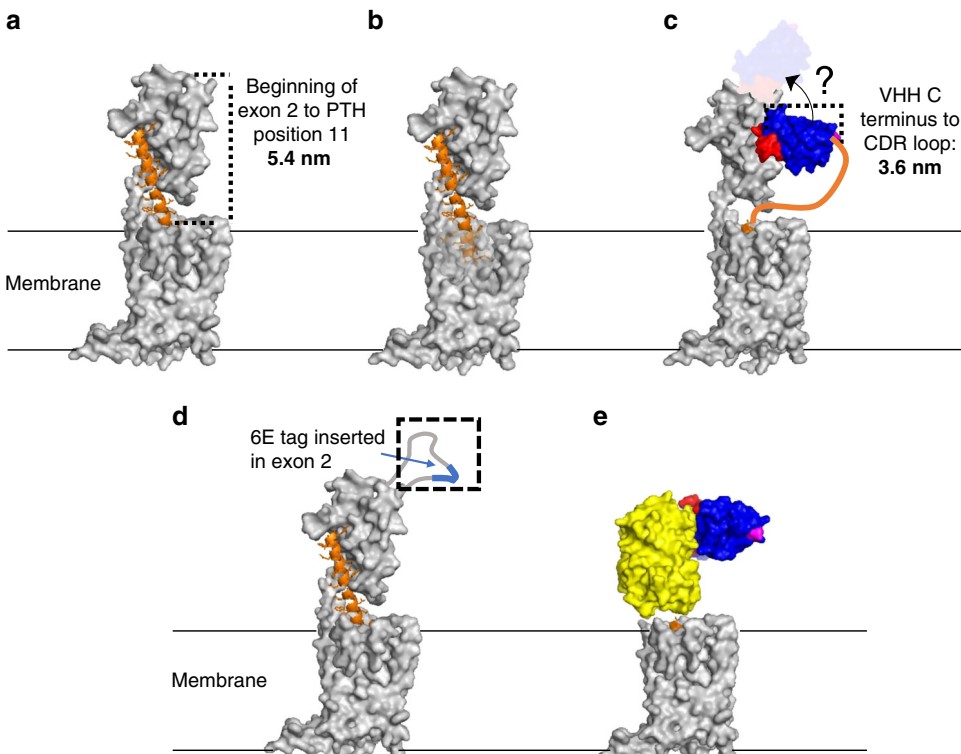

**Fig. 1 Schematic of VHH-mediated delivery of PTHR ligands. a** Crystal structure of human PTHR1 (silver) bound to PTH(1-34) (orange) protein data bank entry (PDB): 6FJ3. **b** Structure as in panel a but with PTHR1 residues 231–296 and 349–353 shown in transparency to allow visualization of the N-terminal portion of PTH inserted into the transmembrane domain. **c** Modeled complex of PTHR1 with VHH-PTH(1-11). The VHH structure (blue) is based on VHH$_{GFP}$ from PDB: 3K1K and PTH(1-11) bound to receptor (orange) is derived from PDB: 6FJ3. The VHH is colored blue except for the complementarity determining loops (red), which bind the target, and the C terminus (magenta), where the PTH fragment is attached. Neither the site of binding for VHH$_{PTHR}$, nor its orientation relative to PTHR1 is known, as indicated by the ghost version of the VHH. Modeled structures of **d** PTHR1$_{6E}$, **e** PTHR1$_{YFP\Delta ECD}$. **d** The predicted location of the PTHR1 segment encoded by exon 2 is highlighted in the dashed box. The orientations of the inserted tags (6E-blue, YFP-yellow) relative to the remainder of the receptor are not known. **e** Residues 31–179 from PTHR1 and residues 12–34 from PTH (PDB: 6FJ3) were removed to provide this structure. PTHR1$_{YFP\Delta ECD}$ is depicted in complex with VHH$_{GFP}$ (blue; PDB 3K1K).

suggest that CLAMPs should be broadly applicable for the design of ligands with unique and useful properties.

## Results

**Receptor constructs and conjugates used for targeting.** PTH(1-34) interacts with PTHR1 via a two-site mechanism of interaction (Fig. 1a, b)[8,16]. The association between the extracellular domain of PTHR1 and residues 15–34 of PTH provides the bulk of the binding energy and specificity for this interaction. The association between the transmembrane domain of PTHR1 and residues 1–14 of PTH induces a conformational change in the receptor, which initiates intracellular signaling cascades. This mode of interaction, supported by a large amount of structure-activity relationship data, has been confirmed recently by crystallography and cryo-electron microscopy of PTHR1-ligand complexes (Fig. 1a)[13,14].

To mimic receptor association exhibited by PTH(1-34), we used either wild-type PTHR1 or PTHR1 variants modified to carry an epitope in the extracellular domain recognized by a VHH of choice. While there is no structural information for any VHH bound to PTHR1, we envisioned a mode of interaction between the receptor and VHH-PTH conjugates like that depicted in Fig. 1c. The portion of PTHR1 encoded by exon 2 is not resolved in structural studies (Fig. 1d)[13,14,17], is not important for ligand binding[18], and in past work has been targeted as a site for receptor modification[18,19]. We generated a construct that encodes a PTHR1 variant in which a 14-residue fragment from exon 2 was replaced with a 14-residue epitope tag from the intracellular protein UBC6e (PTHR1$_{6E}$, Fig. 1d)[20]. We also used a receptor construct in which a pH-sensitive green fluorescent protein variant (GFP) was inserted into the portion of the receptor encoded by exon 2 (PTHR1$_{GFP}$ in Supplementary Fig. 1)[19]. Another version of PTHR1 in which yellow fluorescent protein (YFP) replaces the entire N-terminal extracellular domain (PTHR1$_{YFP\Delta ECD}$, Fig. 1e)[21], was also used.

To target these receptors, we constructed conjugates comprised of N-terminal fragments of PTH and VHHs (Fig. 1). We used VHHs that recognize green or yellow fluorescent proteins (VHH$_{GFP}$)[22], a 14-mer peptide fragment from the intracellular protein UBC6e (VHH$_{6E}$)[20], or PTHR1 itself (VHH$_{PTHR}$)[23]. We expressed C-terminally His-tagged VHHs in bacteria in a form amenable to subsequent site-specific functionalization at the C-terminus, using sortase A-mediated labeling (sortagging)[24,25]. To these purified VHHs we attached either a triglycine-modified fluorophore for cytofluorimetry or a peptide with azide and biotin moieties for biorthogonal chemistry and conjugate tracking, respectively (Fig. 2).

We determined whether VHHs would bind to their intended targets on live cells by flow cytometry (Fig. 3a). We stained HEK293 cell lines stably transfected with the PTHR1 variants described above, rat PTHR1 (rPTHR1)[26], or PTHR2. rPTHR1 has been studied extensively and is identical to murine PTHR1 (mPTHR1) in

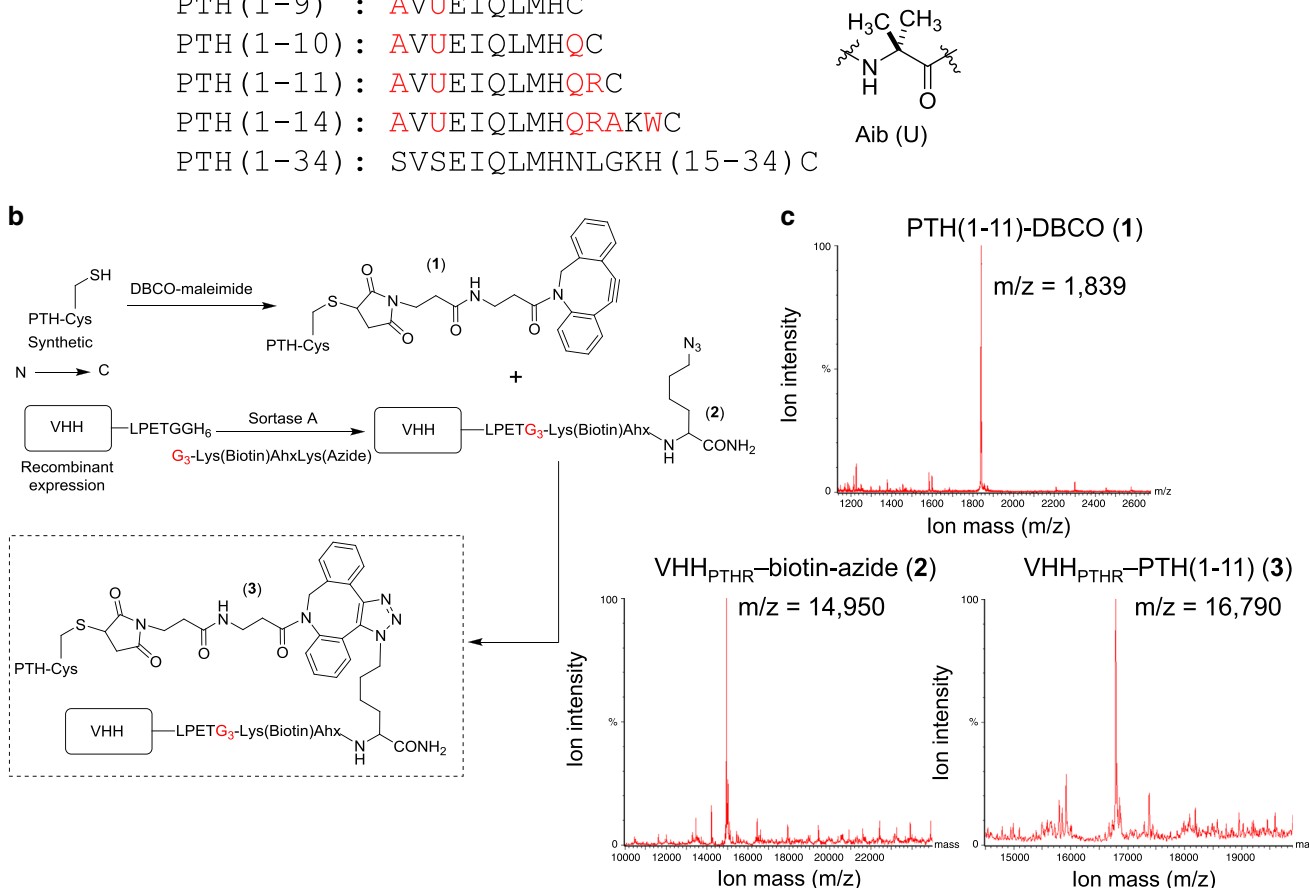

**a**
```
PTH(1-9)  : AVUEIQLMHC
PTH(1-10): AVUEIQLMHQC
PTH(1-11): AVUEIQLMHQRC
PTH(1-14): AVUEIQLMHQRAKWC
PTH(1-34): SVSEIQLMHNLGKH(15-34)C
```

Aib (U)

**b**

PTH-Cys Synthetic — DBCO-maleimide → (1) PTH-Cys

N → C

VHH Recombinant expression — LPETGGH$_6$ — Sortase A / G$_3$-Lys(Biotin)AhxLys(Azide) → VHH — LPETG$_3$-Lys(Biotin)Ahx Lys(Azide) (2) CONH$_2$

+

(3) PTH-Cys — VHH — LPETG$_3$-Lys(Biotin)Ahx — CONH$_2$

**c**
PTH(1-11)-DBCO (1)
m/z = 1,839

VHH$_{PTHR}$–biotin-azide (2)
m/z = 14,950

VHH$_{PTHR}$–PTH(1-11) (3)
m/z = 16,790

**Fig. 2 Synthetic peptides and conjugation strategy. a** Structure of synthetic peptides used in this study. Residues that differ from human PTH and are derived from the M-PTH structural series are shown in red[29]. M-PTH refers to a modified analog developed in past structure-activity relationship studies. The residue denoted "U" corresponds to aminoisobutyric acid (Aib), structure at right. **b** Synthetic scheme used to prepare PTH-VHH C-to-C terminal fusions. **c** Mass spectra from the preparation of VHH$_{PTHR}$-PTH(1-11) conjugates. Complete lists of mass spectral data for peptides and conjugates are found in Supplementary Figs. 2, 3.

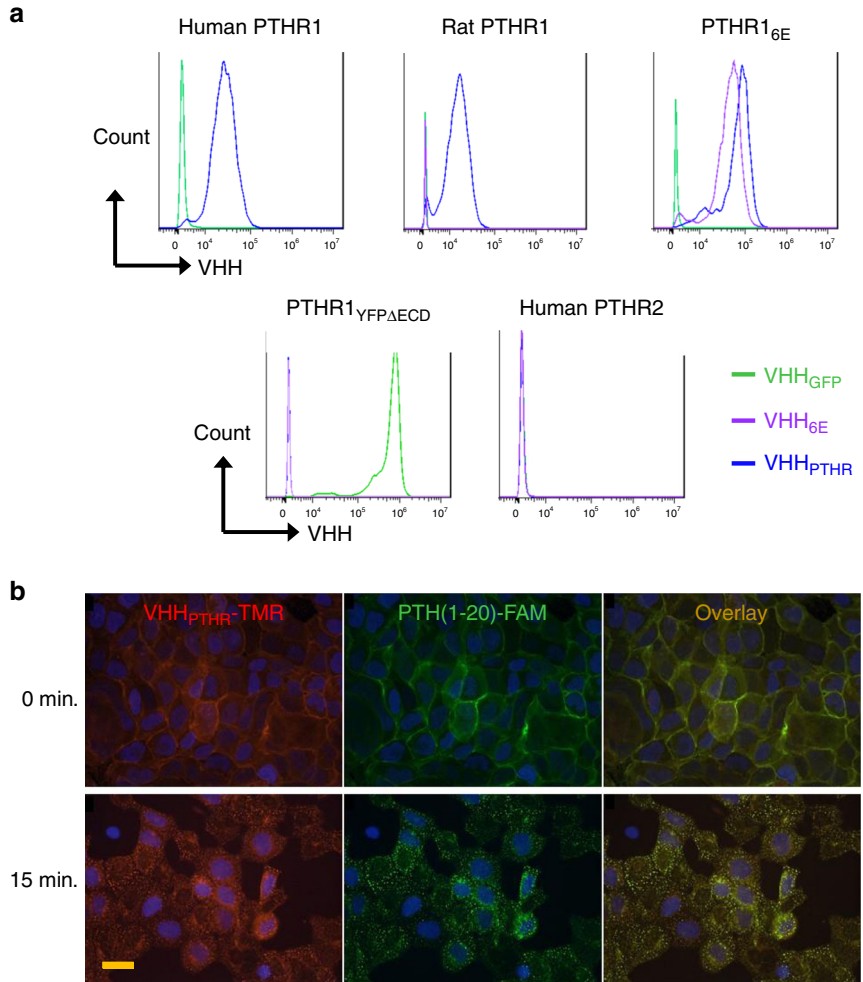

**Fig. 3 Binding of VHHs to HEK293 cell lines stably transfected with PTHRs. a** Analysis of VHH binding to PTHR1, PTHR2 and variants by flow cytometry. HEK293 cell lines in suspension were incubated on ice with 100 nM VHH sortagged with Alexafluor647, pelleted by centrifugation, washed and analyzed. Data for PTHR1-GFP is found in Supplementary Fig. 1. **b** Analysis of VHH binding with microscopy. Adherent HEK293 cells expressing human PTHR1 were stained on ice with 50 nM VHH_PTHR-tetramethylrhodamine (TMR, red) and 30 nM PTH(1-20)-fluorescein (FAM, green) for 30 min. Following staining, cells were washed and treated with fixative in preparation for image acquisition either immediately after staining (0 min) or following a 15-min incubation in medium at room temperature (15 min). Nuclei (blue) were stained with DAPI. The scale bar (20 μm) is at the bottom left image and is applicable to each image in this panel. This experiment was repeated twice with similar results.

the extracellular domain outside of exon 2. VHHs that bind to rPTHR1 should also bind to mPTHR1 and be useful for studies in mice. Each of the VHHs stained cell lines as expected, with the exception of the VHH_GFP-PTHR1_GFP pair, as discussed in Supplementary Fig. 1. VHH_PTHR bound all constructs that retained the PTHR1 ECD, including rat PTHR1, but not PTHR1_YFPΔECD. This places the binding site of VHH_PTHR in the PTHR1 extracellular domain (ECD). Only VHH_GFP stained cells that express PTHR1_YFPΔECD, consistent with its ability to bind YFP[22]. None of the VHHs tested stained the cell line that expresses PTHR2[27]. To estimate the affinity of the selected VHH for their targets we used flow cytometry to measure binding. The staining of PTHR1_YFPΔECD by VHH_GFP and PTHR1_6E by VHH_6E exhibited half-maximal staining at <10 nM, whereas the staining of cells expressing each of the PTHR1 receptor constructs that retained the ECD reached half-maximal staining at 100–200 nM (Supplementary Fig. 4). The precise half-maximal staining concentrations for VHH_PTHR are unknown because the intensity of staining (MFI) did not plateau at the highest concentrations tested and binding strength was estimated from other experiments.

We used microscopy to complement flow cytometry and visualized distribution of either PTHR1 or PTHR1_6E following engagement by PTH(1-20), functionalized with fluorescein, and either VHH_PTHR or VHH_6E tagged with tetramethylrhodamine (Fig. 3b, Supplementary Fig. 5). Imaging of cells fixed immediately after staining on ice shows colocalization of VHH and PTH at the cell surface. Following a 15-min incubation at room temperature, we observed punctate and colocalized fluorescent signals, corresponding to endocytosed receptor-PTH-VHH complexes[28]. Cells not transfected with PTHR1 showed weak staining (Supplementary Fig. 5). These data indicate that the indicated VHHs and PTH (1-20) can simultaneously engage the receptor.

**PTHR1 peptide ligands and conjugation to antibody fragments.** To test whether delivery of PTH fragments to their site of action by conjugation to VHHs affects their signaling activity, we synthesized N-terminal fragments of PTH (Fig. 2, Table 1, Supplementary Fig. 6). These fragments were prepared as C-terminal amides by conventional solid-phase peptide synthesis, and purified. Their identities were confirmed by mass spectrometry

**Table 1 Stimulation of PTHR1 and variants by VHH-PTH conjugates.**

| Peptide or conjugate | hPTHR1 | | | $PTHR1_{6E}$ | | | $hPTHR1_{YFP\Delta ECD}$ | | |
|---|---|---|---|---|---|---|---|---|---|
| PTH(1-34) | 0.51 | ± | 0.28 | 1.3 | ± | 1.0 | 689 | ± | 301 |
| PTH(1-14) | 4.3 | ± | 2.0 | 3.4 | ± | 1.6 | 1.1 | ± | 0.9 |
| PTH(1-11) | 516 | ± | 238 | 94 | ± | 74 | 246 | ± | 133 |
| PTH(1-10) | 3121 | ± | 1671 | 5079 | ± | 407 | 3841 | ± | 1604 |
| PTH(1-9) | | Inactive | | | Inactive | | | Inactive | |
| $VHH_{PTHR}$-PTH(1-14) | 0.075 | ± | 0.041 | 0.2 | ± | 0.1 | 0.9 | ± | 0.5 |
| $VHH_{PTHR}$-PTH(1-11) | 5.0 | ± | 1.6 | 4.0 | ± | 3.2 | | Inactive | |
| $VHH_{PTHR}$-PTH(1-10) | | Inactive | | | ND | | | ND | |
| $VHH_{6E}$-PTH(1-14) | | Inactive[a] | | 0.4 | ± | 0.2 | | ND | |
| $VHH_{6E}$-PTH(1-11) | | Inactive | | 6.9 | ± | 2.6 | | Inactive | |
| $VHH_{6E}$-PTH(1-10) | | ND | | 2.8 | ± | 1.4 | | ND | |
| $VHH_{6E}$-PTH(1-9) | | ND | | | Inactive | | | ND | |
| $VHH_{GFP}$-PTH(1-14) | | Inactive | | | Inactive | | 0.58 | ± | 0.29 |
| $VHH_{GFP}$-PTH(1-11) | | Inactive | | | Inactive | | 0.14 | ± | 0.06 |
| $VHH_{GFP}$-PTH(1-10) | | ND | | | ND | | 0.46 | ± | 0.22 |
| $VHH_{GFP}$-PTH(1-9) | | ND | | | ND | | | ~40% active[b] | |

HEK293 cell lines were treated with varied doses of the indicated peptides or conjugates. Activation was assessed by measuring luminescence from a cAMP-activated luciferase variant. Values listed represent EC50 values (mean ± SD, nM). Each value comes from ≥3 independent experiments. Further details, including the number of replicates for each measurement and the normalized maximal responses induced, are reported in Supplementary Table 1. ND indicates that the measurement was not made. Inactive indicates that the luminescence response measured at the highest concentration tested (100 nM for VHH-peptide conjugates, 10,000 nM for peptides) was <5% of the maximal response induced for that cell line.
[a]The highest concentration tested for this conjugate was 320 nM.
[b]The precise EC50 value could not be obtained but at the highest concentration tested (100 nM) this compound induced a response ~40% that of the maximal response observed. Data for PTHR1-GFP are shown in Supplementary Fig. 1. Representative dose-response curves are shown in Supplementary Fig. 6.

(Fig. 2, Supplementary Fig. 2). Most of these peptides contained several of the modifications found in the M-PTH series of PTH peptides, including the non-standard residue aminoisobutyric acid (Aib) at position 3, which enhances the biological activity of these short PTH fragments (Fig. 2)[29]. Each of these peptides contained a C-terminal cysteine (Cys). Using Cys-maleimide chemistry we appended a dibenzylcyclooctyne (DBCO) handle (Supplementary Fig. 2) to enable an azide-alkyne conjugation between the C-termini of an azide-functionalized VHH and a DBCO-modified synthetic peptide. Of note, the resulting triazole linkage is not susceptible to cleavage by reduction, unlike the disulfide linkages used in other conjugates. The composition of the conjugates was confirmed by mass spectrometry (Fig. 2, Supplementary Fig. 3). For comparison, we also prepared conjugates in which a PTH(1-14) analog with an N-terminal triglycine extension ($G_3$-PTH(1-14)) was conjugated to VHHs using sortagging, resulting in a conjugate with the more conventional C-N configuration (Supplementary Fig. 7).

We then assessed the capacity of these peptides and conjugates to stimulate the production of cyclic adenosine monophosphate (cAMP), a second messenger molecule produced upon PTHR1 activation, using HEK293 cells expressing a PTHR variant targeted by the relevant VHH and a luciferase-based cAMP-responsive reporter[30]. Progressive truncation of C-terminal residues from PTH(1-34) caused a marked loss in the potency on wild-type PTHR1 and other PTHR1 variants with intact ECDs (Table 1, Supplementary Fig. 6). Addition of a triglycine appendage at the N-terminus of PTH(1-14) caused a reduction in potency, relative to PTH(1-14) with a free N-terminal amine (Supplementary Fig. 7), in line with precedent[31]. Conjugates in which $G_3$-PTH(1-14) was ligated to the VHH C-terminus using sortase were completely inactive, emphasizing the importance of a free N-terminus for PTH and its fragments (Supplementary Fig. 7). In contrast, conjugates formed by C-to-C-terminal fusion were active (Table 1, Supplementary Fig. 6).

The conjugation of PTH fragments lacking residues 15–34, known to be important for ECD binding, to VHHs that bound to the targeted receptor showed a strong increase in potency (Table 1). For example, $VHH_{GFP}$-PTH(1-10) is 7800-fold more potent than PTH(1-10) on $PTHR1_{YFP\Delta ECD}$. The potency of

$VHH_{GFP}$-PTH(1-10) at $PTHR1_{YFP\Delta ECD}$ ($EC_{50} \sim 0.5$ nM) is especially notable given that PTH(1-34), an analog with properties similar to naturally occurring PTH, is relatively weakly active ($EC_{50} > 500$ nM) on this receptor. $VHH_{PTHR}$ conjugation also increases the potency of PTH(1-11) and PTH(1-14) at PTHR1, $PTHR1_{GFP}$, and $PTHR1_{6E}$, in line with results from VHH binding experiments (Supplementary Fig. 4). As an example, $VHH_{PTHR}$-PTH(1-14) is 57-fold more potent than PTH(1-14) on cells that express wild-type PTHR1. Even VHHs that showed weak staining of the relevant cell lines in cytofluorimetry, like that of $VHH_{GFP}$ on $PTHR_{GFP}$-expressing cells, still enhanced the signaling activity of N-terminal fragments like PTH(1-11) (Supplementary Fig. 1). Signaling duration, as assessed using a previously validated method[27,29,32], is also prolonged for the shorter PTH fragments when conjugated to the appropriately specific VHH (Supplementary Fig. 8). The kinetics of cAMP signaling induced by some VHH-PTH fragments resemble that seen with PTH(1-34) (Supplementary Fig. 8). The ability of a ligand to induce prolonged signaling at PTHR1 is correlated with continued signaling upon internalization into endosomal compartments, PTH(1-34) serving as a prime example[29]. The prolonged signaling of VHH-PTH conjugates relative to the corresponding free peptides suggests that the added affinity provided by VHH binding may enable endosomal signaling.

In contrast to the increase in signaling activity provided by the conjugation of PTH fragments to receptor-binding VHHs, conjugation of active PTH fragments with irrelevant VHHs is detrimental to activity on the intact PTHR1. For example, even at the highest concentrations tested, conjugates of $VHH_{6E}$ and $VHH_{GFP}$ with PTH(1-11) and PTH(1-14) are inactive on wild-type human PTHR1, even though the peptides themselves are quite active (Table 1, Supplementary Fig. 6). This loss of activity is caused at least in part by a loss in receptor binding for PTH fragments conjugated to irrelevant VHHs (Supplementary Fig. 9). $VHH_{6E}$-PTH(1-14) fails to bind hPTHR1 expressing cells, whereas $VHH_{PTHR}$-PTH(1-14) binds more tightly than $VHH_{PTHR}$ alone. The impact of irrelevant VHH conjugation is not explained by variation in signaling activity caused by installation of the 'click' handles (Supplementary Fig. 10). Furthermore, the length of the VHH-PTH linker is not a strong determinant of conjugate

signaling activity or specificity: incorporation of a $PEG_3$ linker has minimal impact (Supplementary Fig. 10). The enhanced signaling activity provided by VHH conjugation is not seen with PTHR1 ligands that bind through both ECD and transmembrane domain interactions: conjugation of PTH(1-34) with VHHs yields active compounds, regardless of whether the target of the VHH is present on the cell line tested (Supplementary Fig. 11). $VHH_{PTHR}$-PTH(1-14) activated $PTHR1_{YFP\Delta ECD}$, even though the VHH does not bind to this receptor (Table 1).

Among the GPCR superfamily, family B GPCRs have relatively large ECDs. To assess whether the CLAMP approach might also be useful for GPCRs with smaller ECDs, we targeted a variant of PTHR1 in which the entire ECD is replaced by the 6E epitope tag (QADQEAKELARQIS, Supplementary Fig. 12)[20]. Application of $VHH_{6E}$-PTH(1-11) activated PTHR1-delNT-6E more effectively than PTH(1-11) (Supplementary Fig. 12b). In control experiments, PTHR1-delNT (no 6E tag) was not activated by $VHH_{6E}$-PTH(1-11) (Supplementary Fig. 12c).

Activated GPCRs can signal through more than one intracellular pathway, at the cell surface or from cell-internal compartments. There is interest in identifying ligands that are functionally selective in signaling through one pathway over another (biased agonists)[33]. PTHR1 signals through multiple pathways, including Gs/protein kinase A (PKA)/cAMP, Gq/phospholipase C (PLC)/ $Ca^{2+}$, and β-arrestin/ERK[34]. We tested whether selected VHH-PTH conjugates engaged these pathways. $VHH_{PTHR}$-PTH(1-14) but not $VHH_{PTHR}$-PTH(1-11) stimulated signaling though the Gq/PLC/ $Ca^{2+}$ signaling pathway in cells that express PTHR1 (Supplementary Fig. 13). $VHH_{PTHR}$-PTH(1-11) appears to be selective for Gs/PKA/cAMP signaling, although assessing Gq signaling at higher conjugate concentrations than currently possible or using different assay formats might yet reveal weak activity. PTHR1 signaling through the Gq is more sensitive to structural modifications and alterations in affinity than signaling through the Gs pathway[8,34], in line with these findings. We also determined whether $VHH_{PTHR}$-PTH(1-14) could induce PTHR1 to recruit β-arrestin. PTH(1-34) and $VHH_{PTHR}$-PTH(1-14) (Supplementary Fig. 14a, d), but not $VHH_{PTHR}$ alone (Supplementary Fig. 14e), stimulated the relocalization of cytoplasmically disposed YFP-tagged β-arrestin into distinct puncta. Colocalization of a fluorophore-tagged PTH(1-34) with YFP-β-arrestin in puncta supported the specificity of arrestin recruitment to the agonist-occupied PTHR1 (Supplementary Fig. 14c). In a separate experiment, cells that express YFP-β-arrestin, transfected with a PTHR1 construct containing an HA-epitope tag in exon 2 (PTHR1-HA), were treated with $VHH_{PTHR}$-PTH(1-14). This resulted in punctate YFP signals that colocalized with PTHR1-HA (Supplementary Fig. 15a–c). Many of the puncta observed in cells treated with PTH(1-34) or $VHH_{PTHR}$-PTH(1-14) were observed near the nucleus (Supplementary Figs. 14, 15), consistent with ligand-induced internalization. No colocalized puncta were observed in cells treated with $VHH_{PTHR}$, and HA-staining appeared restricted to the cell surface (Supplementary Fig. 15d–f). We further examined ligand-induced internalization through the use of cells expressing PTHR1-GFP, in which the GFP variant is pH-sensitive. Since the spectral properties of this GFP variant change as a function of pH[19], the movement of the receptor from the cell surface into the acidic endolysosomal compartment can be followed by monitoring the change in wavelength of emitted fluorescence after addition of ligand (Supplementary Fig. 16). In this assay, $VHH_{PTHR}$-PTH(1-14) behaves similarly to PTH(1-34), an agonist known to induce PTHR1 internalization[34], providing further evidence that $VHH_{PTHR}$-PTH(1-14) induces internalization. Overall, $VHH_{PTHR}$-PTH(1-14) behaves similarly to PTH(1-34) in each of these cell-based assays tested.

The selectivity of PTH fragments for tagged receptors imparted through conjugation to VHHs led us to test whether similarly enhanced selectivity could be achieved between two naturally occurring subtypes of PTHR. PTH(1-34) tightly binds and activates both PTHR1 and PTHR2[8]. $VHH_{PTHR}$ binds to PTHR1 but not PTHR2 (Fig. 3). $VHH_{PTHR}$-PTH conjugates should therefore activate PTHR1 but not PTHR2. We focused on conjugates of PTH(1-14), as this fragment also activated PTHR1 and PTHR2 (Fig. 4). The $VHH_{PTHR}$-PTH(1-14) conjugate activated PTHR1 more potently than any other compound tested in this study ($EC_{50}$~0.07 nM), whereas it was completely inactive at PTHR2 at 330 nM (>4500-fold selectivity for PTHR1, Fig. 4). This contrasts with the observed lack of selectivity of PTH(1-34) (5-fold selectivity for PTHR1; here and in past work[8]).

**In vivo activity**. We tested whether the potent biological activity observed for VHH-PTH conjugates in cell-based assays would extend to an in vivo setting. We used the $VHH_{PTHR}$-PTH(1-14) conjugate in these experiments because it was more potent than $VHH_{PTHR}$-PTH(1-11). $VHH_{PTHR}$ potentiated PTH(1-11) signaling activity for the rat PTHR1 (Supplementary Fig. 17). Since $VHH_{PTHR}$ bound rat PTHR1 (Fig. 3) we were confident that it would also bind mouse PTHR1, as these receptors are 99% identical in their extracellular domain. To measure in vivo activity, we injected mice subcutaneously with equimolar amounts of either PTH(1-34), M-PTH(1-14), $VHH_{PTHR}$-PTH(1-14), or saline. Doses were chosen based on precedent for the treatment of mice with PTH(1-34) to stimulate acute calcemic responses[29,32]. PTH(1-34) induced a strong increase in blood ionized calcium levels, which peaks 1-2 h after injection and returns to baseline thereafter, whereas free M-PTH(1-14) exhibits little if any activity in this assay, in line with past findings[29]. Our experiment showed that $VHH_{PTHR}$-PTH(1-14) stimulated a spike in blood calcium that peaked two hours after injection (Fig. 5). The conjugation of PTH(1-14) with $VHH_{PTHR}$ therefore potentiates biological activity both in cell-based assays and in vivo.

## Discussion

Antibodies as part of conventional antibody-drug conjugates deliver cytotoxic compounds that typically target intracellular proteins[1,2]. Less explored is the use of antibodies to deliver ligands for surface receptors such as GPCRs. This is likely due to complications in preparing homogenous and bioactive conjugates of antibodies and ligands that rely on antibody binding to potentiate engagement of receptor by the ligand. Immunocytokines (conjugates of cytokines and antibodies) are an exception[35]. Immunocytokines have progressed to the clinic, but not without toxicity, suggesting an inadequacy in targeting[35]. In one case, even the identity of the targeting antibody of the immunocytokine was irrelevant for its in vivo efficacy[36]. Mutations in cytokines, introduced to improve the selectivity of immunocytokines, can dampen the affinity for their receptors[37,38]. Immunocytokine-based approaches differ from the CLAMP platform in that they rely on genetic fusions (and genetically encoded residues) and the use of full-size cytokine domains, as opposed to the small peptide fragments with non-natural residues used here.

In one precedent for targeting GPCRs with antibody-ligand fusions, $VHH_{GFP}$ equipped with a SNAP-tag was linked to a photoactivatable ligand for the GPCR mGluR2[39]. This fusion was then used to activate a GFP-tagged receptor upon photoactivation of the ligand. The response induced by a saturating concentration of the photoactivatable VHH-ligand conjugate was ~40% of that induced by a saturating dose of natural ligand and required the

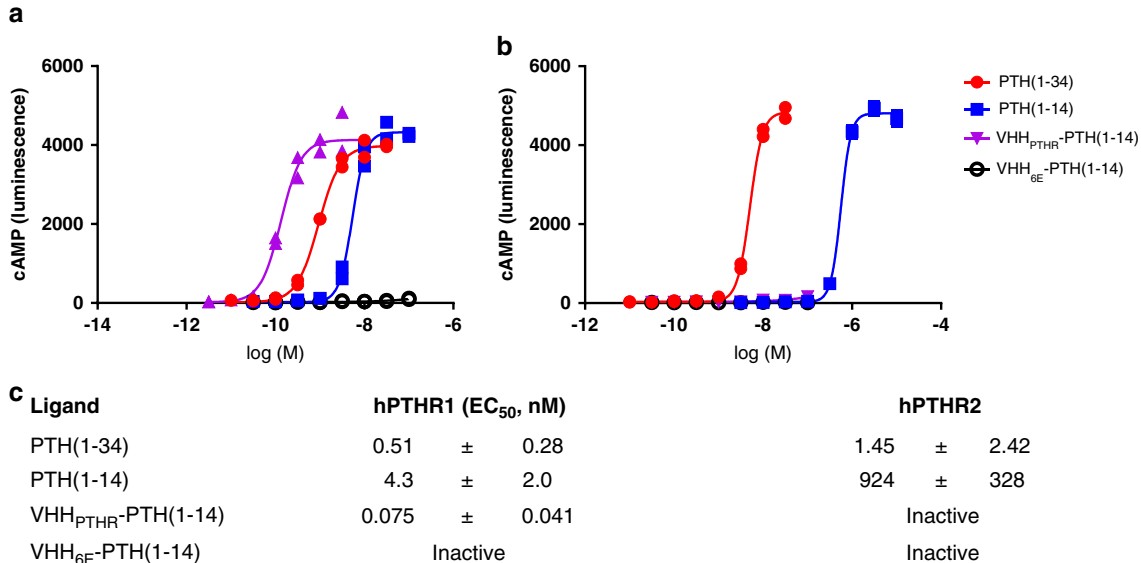

**Fig. 4 Selective and potent activation of PTHR1 via VHH$_{PTHR}$ conjugation.** HEK293 cell lines stably expressing either human PTHR1 (hPTHR1) or hPTHR2 were treated with varied doses of the indicated peptides or conjugates and activation was assessed by cAMP production. **a, b** Representative dose-response curves from a single experiment for **a** hPTHR1 or **b** hPTHR2 activation. Each data point corresponds to readings from a single well in the indicated experiment ($n = 2$ data points per condition). Full descriptions of data from multiple experiments are found in Supplementary Table 1. Curves result from fitting of a sigmoidal dose-response model to data. 'Inactive' indicates that the response induced at the highest concentration tested was <5% of the maximal response induced for that cell line. Red circles correspond to responses induced by PTH(1-34); blue squares to PTH(1-14); purple triangles to VHH$_{PTHR}$-PTH(1-14); and black open circles to VHH$_{6E}$-PTH(1-14). **c** Tabulation of cAMP induction potencies. Data for hPTHR1 are identical to those in Table 1 and are included here for comparison. Values listed represent EC$_{50}$ values (mean ± SD). Each value comes the following number of independent experiments: hPTHR1-PTH(1-34), $n = 8$; hPTHR1-PTH(1-14), $n = 5$; hPTHR1-PTH(1-34), $n = 4$; hPTHR1-VHH$_{PTHR}$-PTH(1-14), hPTHR1-VHH$_{6E}$-PTH(1-14), hPTHR2-PTH(1-14), hPTHR2-VHH$_{PTHR}$-PTH(1-14), and hPTHR2-VHH$_{6E}$-PTH(1-14) $n = 3$. Further details, including the relative levels of maximal responses, are reported in Supplementary Table 1. Note that the x-axes in these graphs differ as peptides exhibit weaker activity for PTHR2. Source data can be found in the Source Data file.

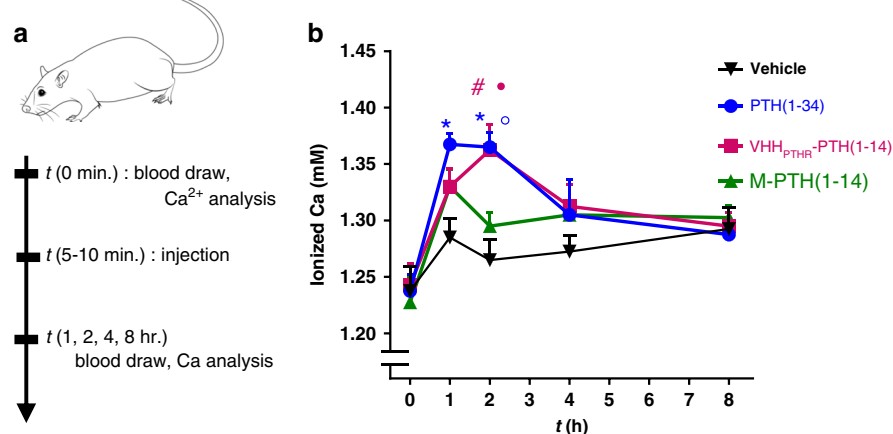

**Fig. 5 VHH conjugation potentiates an in vivo response. a** Schematic of the experiment performed in mice. **b** Measurement of blood ionized calcium levels in mice injected with PTH and conjugates. Data presented is from a single experiment. The double line break represents a discontinuity in the Y-axis. Mice (CD1 females, 11 weeks) were injected subcutaneously with the indicated ligand (dose = 35 nmol/kg). Blood was drawn at the indicated time points and analyzed for ionized calcium levels. Data points indicate mean ± standard error of the mean (SEM), $n = 4$. A two-sided t-test without corrections for multiple comparisons was used for comparisons. *$p = 0.005$ vs. vehicle. #$p = 0.015$ vs. vehicle. ●$p = 0.038$ vs. M-PTH(1-14). °$p = 0.008$ vs. M-PTH(1-14). The sequence of M-PTH (1-14) used here differs from PTH(1-14) in Fig. 2 and is UVUEIQLMHQXAKW where U is Aib and X is homoarginine. Source data are provided in the Source Data File. Black triangles correspond to data for vehicle; blue circles to PTH(1-34); magenta squares to VHH$_{PTHR}$-PTH(1-14); and green triangles to M-PTH(1-14). Traces for the calcemic responses of individual mice is found in Supplementary Fig. 18. Source data can be found in the Source Data file.

use of a receptor-GFP fusion[39]. This strategy precludes the use of genetically unmodified cells or animals. In another example, full-length anti-PCSK9 antibodies fused at the N-terminus of the heavy or light chain with analogs of the glucagon-like peptide 1 (GLP-1) were produced[40]. However, most of the fusions tested were expressed in low yield, isolated with inactivating truncations in the GLP-1 fragment, unstable in solution, or were degraded rapidly in vivo, demonstrating the difficulties encountered when expressing fusion proteins comprised of full-length antibodies and ligands of interest[40].

Despite several screening campaigns, no VHHs that directly activate GPCRs have been identified as of yet[5]. We prepared a library of C-to-C terminal fusions of VHHs and synthetic PTH peptides. The use of C-to-C fusions is supported by the lack of activity of the corresponding N-to-C fusions (Supplementary Fig. 7). It is possible that the genetic fusion of PTH peptides to the N-terminus of VHHs might be accommodated with retention of both VHH binding and PTH activity. This would require a unique genetic construct and optimization of expression for each fusion. We avoid these drawbacks through our chemoenzymatic approach. A further benefit of our synthetic strategy compared to a purely genetic approach is the ease of incorporation of non-natural residues such as Aib into the peptide portion of the conjugate to improve proteolytic stability[40]. Several conjugates stimulated cAMP responses with potencies similar to that of PTH (1-34) (Table 1). Even PTH(1-9), which fails to activate PTHR1 unless tethered directly to the receptor's N-terminus via genetic fusion[41], showed activity when conjugated to a VHH (Table 1). VHH-mediated delivery of ligands should enable identification of weak ligands that might otherwise be dismissed as completely inactive. Similar observations were made in evaluating conjugates consisting of peptide fragments derived from the N- and C-termini of corticotrophin releasing factor-1, which were weakly active or inactive alone, but once assembled via click chemistry, several conjugates were potent agonists[42].

Agonist activity for VHH-PTH conjugates was completely dependent on binding of the VHH to the receptor being targeted: a mismatch between specificity of the VHH and the receptor construct led to a loss in conjugate activity. We identified a conjugate, $VHH_{PTHR}$-PTH(1-14), with very potent signaling activity in cell-based assays (Table 1), with biological activity in mice (Fig. 5), and with selectivity for PTHR1 over PTHR2 that far surpasses the selectivity of PTH(1-34), the prototypical PTHR1 agonist (Fig. 4) used clinically. PTHR1 mediates the biological activity of PTH in treating osteoporosis, whereas the function of PTHR2 is more obscure. Tools to selectively target PTHR1, and subtypes of GPCRs in other families, will be useful for dissecting the biological function of receptors for which potent and selective ligands are scarce. Success in targeting PTHR1 over PTHR2 sets the stage for designing ligands that specifically activate other receptors with overlapping specificities[6,7].

The ability to deliver ligands to specific subtypes of receptors, or to receptors engineered to contain an antibody-recognized tag, should allow the creation of (modular) versions of designed receptors exclusively activated by designer drugs (DREADDs)[43]. Previously described DREADDs for GPCRs were identified through modification of the ligand binding site of naturally occurring GPCRs, so that the modified receptors respond to a designer small molecule but not the ligand of the prototype receptor. These designer molecules selectively activate the designer receptor but not any endogenously expressed alternative[44,45]. A similar approach has been deployed to produce an orthogonal receptor-ligand pair for interleukin-2[46]. Our finding that $VHH_{GFP}$-PTH(1-11) potently activates PTHR1$_{YFP\Delta ECD}$ (EC50 ~ 0.15 nM) but is inactive at wild-type PTHR1, suggests a path toward using VHH-tag recognition as a way to convert a GPCR of choice into a DREADD. One aspect of GPCR pharmacology that has not been faithfully reproduced in some DREADD constructs is that of ligand binding kinetics[47]. For some receptors, such as PTHR1, the duration of ligand binding and the signaling induced as a consequence can dictate the type of physiological response evoked. The duration of the cAMP response elicited by PTHR1 activation is correlated with the strength and duration of the calcemic response in vivo[29,48]. Several of the VHH-PTH conjugates tested here induce cAMP signaling that is prolonged relative to the free peptide and similar

to that of PTH(1-34) (Supplementary Fig. 8), suggesting that the affinity provided by VHH binding can be used as an independent means to adjust ligand binding and signaling kinetics.

In conclusion, we show that the conjugation of otherwise suboptimal PTHR1 agonist peptides to VHHs that target the intended receptor provides a substantial increase in agonist potency and receptor selectivity. The ability to modulate receptor affinity while not modifying the structure of the agonist used to activate signaling should enable a further dissection of connections between ligand affinity, receptor signaling kinetics, and ligand bias[49]. Preliminary analyses suggest that VHH-ligand conjugates can be designed that possess signaling properties that diverge from that of the natural ligands (Table 1, Supplementary Fig. 13). The CLAMP platform should be amenable to targeting other GPCRs, especially those with large peptide ligands that bind to their receptors via a two-site mechanism, such as family B GPCRs and chemokine receptors. Efforts are underway to expand this platform to other GPCR-ligand systems. The applicability of this platform will expand as VHHs that bind to new targets on the cell surface are discovered[5].

## Methods

**General.** HEK293 cell lines (ATCC) were cultured in DMEM medium supplemented with 10% (v/v) fetal bovine serum and penicillin/streptomycin. Cell lines were routinely tested for mycoplasma infection. LC/MS was performed on a Waters Xevo Q-Tof system equipped with HPLC-C8 columns. Mass spectra were obtained using Q-Tof mass spectrometry with a positive ionization mode. Masses for VHHs and conjugates were calculated via analysis of multiply charged ions using the MaxEnt feature on MassLynx software. Protein and peptide concentrations were calculated using absorption at 280 nm for VHHs and peptides with tryptophan (Trp) residues. For peptides without Trp, the amount of peptide was quantified gravimetrically assuming that the weighed mass consisted of 50% peptide (w/w). Antibody used for staining PTHR1-HA (anti-HA-AF594) was purchased from Biolegend (BioLegend #901511) and used at a final concentration of 20 nM. Transfections of HEK293 were performed using Lipofectamine2000 using manufacturer instructions.

**Plasmids and DNA.** HEK293-derived cell lines stably expressing human PTHR1 (GP2.3), rat PTHR1 (GR35), PTHR1$_{GFP}$ (GPG10), and PTHR1$_{YFP\Delta ECD}$ (GD5Y) along with a cAMP-responsive luciferase variant have been previously reported[19,21,26]. A HEK293 cell line stably expressing β-Arrestin2-YFP (GBR24) was constructed similarly[50]. PTHR1$_{6E}$ was produced using the Q5 Site-directed mutagenesis kit (NEB) and used to prepare a stably transfected HEK293-derived cell line. Annotated sequence data for all PTHR1 constructs are found in Supplementary Fig. 19. Aligned sequences of VHHs used in this study are shown in Supplementary Fig. 20.

**Peptide synthesis.** Peptides were prepared using conventional solid-phase synthesis methods with Fmoc-protection of backbone amines. Synthesis was performed on Rink-amide linker resin to yield C-terminal amides. Backbone deprotection was performed via treatment with piperidine in dimethylformamide (DMF, 20% vol/vol) for 15 min at room temperature. Coupling was performed using Fmoc-protected amino acids (4 equivalents), N,N,N′,N′-Tetramethyl-O-(1H-benzotriazol-1-yl)uronium hexafluorophosphate (HBTU, 4 equivalents), and diisopropylethylamine (DIPEA, 8 equivalents) in DMF for 45 min at room temperature. Fmoc-Lys(biotin)-OH and Fmoc-Lys(azide)-OH were from used from commercial sources without alteration of the synthetic methods described above. Following completion of synthesis, the resin was dried and deprotection was carried out using a solution of 92.5% trifluoroacetic acid, 5% $H_2O$, and 2.5% TIPS. Peptides were precipitated into diethyl ether, pelleted by centrifugation, dried under a stream of air, purified using reversed-phase C18 HPLC using a water-acetonitrile gradient, and lyophilized. The identity and approximate purity of peptides was confirmed by LC/MS (Supplementary Fig. 1). Purified products were dissolved in water (10 mM stock concentration) and stored at −20 °C.

Purified peptides with C-terminal cysteines were subjected to a reaction with a 2-fold molar excess of either DBCO-maleimide (Click Chemistry Tools) or DBCO-PEG$_3$-maleimide (ConjuProbe) (Supplementary Fig. 10) in solvent with 50% (v/v) dimethylsulfoxide (DMSO) and 50 mM pH 7.4 phosphate buffer and purified by reversed-phase C18 HPLC. The identity of peptides was confirmed by LC/MS (Supplementary Fig. 2). Purified products were dissolved in DMSO (1 mM stock concentration) and stored at −20 °C.

**Protein expression and purification.** The production and purification of VHH$_{GFP}$ (named VHH-enhancer) and VHH$_{6E}$ (named VHH05) has been described

previously[20,22]. The sequence for VHH$_{PTHR}$ was acquired from the literature (named 22A3)[23]. Although several VHHs that bound PTHR1 were reported, we chose 22A3 as it was reported to have the highest affinity[23]. Briefly, VHHs were expressed using the pHEN6 vector. Plasmids coding for PelB-VHH-LPETGG-His$_6$ were transformed into WK6 E. coli using heat shock. Transfected WK6 E. coli were grown in Luria Bertani broth under ampicillin selection at 37 °C until an optical density at 600 nm between 0.6 and 0.8 was reached. Protein expression was induced by the addition of 1 mM IPTG and cells were grown at 30 °C overnight. The bacteria were pelleted by centrifugation and resuspended in TES buffer (50 mM Tris, 650 μM EDTA, 2 M sucrose, 15 mL buffer per liter of culture) to prepare for osmotic shock. After incubating for 2 h at 4 °C, 75 ml distilled H$_2$O was added, and the bacterial suspension was incubated overnight at 4 °C. The bacteria were again pelleted and VHHs were purified from the supernatant by Ni-NTA bead batch purification, followed by buffer exchange. Sortase-A pentamutant was expressed and purified as previously described[3].

**Flow cytometry**. Suspensions of cells in PBS were stained for 1 h on ice in the presence of indicated concentrations of VHH probes functionalized with Alexa-fluor647. Cells were pelleted by centrifugation and washed with PBS prior to analysis by flow cytometry (BD Accuri C6). To select intact cells gating was performed on forward scatter/side scatter profiles for analysis (see Supplementary Fig. 4 for an example of the gating strategy). Data were analyzed using FlowJo version 7.6. The median fluorescent intensity (MFI) of stained cells was used to generate VHH binding dose response curves (Supplementary Fig. 4). For curves that did not reach plateau at the highest concentrations tested, curves were constrained by setting the maximal plateau value equal to that seen when staining that cell line with other VHHs that did achieve a plateau.

**Sortase-mediated labeling (sortagging)**. VHHs were labeled using sortase A pentamutant[25]. Briefly, VHH (20–100 μM) with a C-terminal sortase-recognition motif and His-tag were incubated with GGG-peptide (500 μM) and sortase A pentamutant (10 μM) in Tris-buffered saline (TBS) containing 10 mM CaCl$_2$ overnight at 14 °C. DMSO was added at concentrations up to 20% (vol/vol) in cases where conjugates were prone to aggregation, as previously described[51]. Functionalized VHHs were purified from unreacted VHH and sortase by exposure to nickel-NTA sepharose beads and removal of GGG-peptide by buffer exchange using a 10 kDa molecular weight cutoff spin filter or a PD10 disposable size exclusion column. Purified VHH conjugates were concentrated using 3 kDa spin filter. Since VHH-PTH(1-14) and VHH-G$_3$-PTH(1-14) conjugates were prone to precipitation following concentration this step was avoided or minimized.

**VHH-peptide conjugation reactions**. VHH-biotin-azide conjugates (Fig. 2) were mixed with PTH-DBCO (3-fold molar excess) in TBS with 10% (v/v) glycerol. The reaction was shaken at 22 °C until unreacted VHH-biotin-azide had been completely consumed. The product conjugate was purified from free PTH-DBCO using a PD10 size exclusion column. Product identity was confirmed by LC/MS (Supplementary Fig. 3).

**Microscopy**. Monolayers of HEK293 cells grown on glass cover slips at approximately 80% confluency were washed with Hanks balanced salt solution supplemented with 10 mM HEPES pH 7.4 and 0.1% (w/v) bovine serum albumin (HB). The cells were then incubated with peptide, VHH, or VHH-peptide conjugates in HB at 4 or 22 °C for 30 m. After staining, cells were washed with HB three times, and fixed with 4% formalin either immediately after rinsing or following a 15 m incubation at 22 °C in DMEM + 10% FBS. Indicated slides were permeabolized using 0.5% Triton X100, followed by staining with commercial anti-HA-Alexafluor 594 antibody (20 nM, BioLegend #901511). Cells were then rinsed and mounted with Vector-shield containing DAPI (to visualize nuclei) on glass slides for imaging. Images were acquired using a Nikon Eclipse Ni system with a ×40 PLAN FLUOR 0.75NA DIC M/N2 objective.

**Measurement of cAMP response**. These assays were performed as previously described[26]. Briefly, HEK-293-derived cell lines that stably express the Glosensor cAMP reporter (Promega Corp.)[30] and PTHR1, a PTHR1 variant, or PTHR2 were seeded into white sided 96-well plates (50,000 cells/well) and grown to confluency. Confluent monolayers of cells were pre-incubated with CO$_2$ independent medium containing D-luciferin (0.5 mM) at 37 °C until a stable baseline level of luminescence was established (20 min). Varying concentrations of ligands were then added, and the time course of luminescence response was recorded using BioTek plate reader. The maximal luminescence response (observed 12–16 min after ligand addition) was used to construct dose-response data sets (Table 1, Supplementary Fig. 6).

For the measurement of cAMP signaling duration experiments (Supplementary Fig. 8) were performed as previously described[27]. Cells were treated with ligands at the indicated concentrations for 12 min (ligand-on phase). After this period, the medium in each well was removed and the cells were rinsed twice with CO$_2$-independent medium to remove unbound ligand. After the addition of D-luciferin-containing fresh medium to each well, the luminescence was recorded for an additional 30–40 min using a PerkinElmer Envision plate reader (ligand-off phase).

**Measurement of cytoplasmic calcium mobilization**. The mobilization of Ca$^{2+}$ levels was assessed in the HEK293 cell line stably transfected with human PTHR1. Intracellular Ca$^{2+}$ levels were assessed using a cell-permeant Ca$^{2+}$ sensor, Fura2-AM (Invitrogen). Cells in a black 96-well plate were loaded with Fura2-AM in the presence of Pluronic F-127 for 45 min and then rinsed with Hanks buffered saline solution (HBSS). Following an additional 30-min incubation in HBSS, the plate was analyzed using a PerkinElmer Life Sciences Envision plate reader to monitor fluorescence emission at a wavelength of 510 nm, upon excitation at wavelengths of 340 and 380 nm. The data were recorded at 2-s intervals prior to and after ligand addition. The data were calculated as the ratio of the fluorescence signal obtained with excitation at 340 nm to that obtained with excitation at 380 nm.

**Measurement of internalization using GFP fluorescence**. Receptor internalization was assessed in the HEK293 cell line stably transfected with human PTHR1-pHluorin2-GFP (GPG10)[19]. Confluent monolayers of cells in black walled 96-well plates were incubated in HBSS with bovine serum albumin (0.1% w/v) and HEPES buffer (pH 7.4, 10 mM). Peptides or peptide-VHH conjugates were added and wells were analyzed by recording fluorescence readouts with excitation at 485 or 405 nm and emission at 535 nm. Data were analyzed as a ratio of fluorescence intensity following excitation at 485/405 nm over the course of 90 min.

**Animal experiments and measurement of in vivo response**. Mice (CD1 female, age 11 weeks) were treated in accordance with the ethical guidelines adopted by Massachusetts General Hospital. Calcemic response assays were conducted using cohort sizes comparable to past work[26], which provided data adequate for identifying differences in the time course and magnitude of PTH-induced calcemic responses. Peptides and conjugates were administered at doses that allowed for differentiation between compounds with differing levels of in vivo activity[26]. Statistical analyses were performed assuming Gaussian distribution of data. Mice (n = 4 per compound) were injected subcutaneously with vehicle (10 mM citric acid/ 150 mM NaCl/0.05% Tween-80, pH 5.0) or vehicle containing PTH or conjugate at a dose of 35 nmol/kg body weight. Prior to injection, mice were grouped according to basal blood calcium concentrations to ensure each group possessed similar average (mean) blood ionized calcium levels at t = 0. Blood was withdrawn just before injection (t = 0) or at times thereafter. Tail vein blood was collected and immediately analyzed. Blood Ca$^{2+}$ concentration was measured with a Siemens RapidLab 348 Ca2+/pH analyzer.

**Data calculations**. Data were processed using Microsoft Excel and GraphPad Prism 6. Data from cAMP dose–response assays were analyzed using a sigmoidal dose–response model with variable slope. Data sets were statistically compared by using Student's t test (two-tailed) assuming unequal variances for the two sets.

**Reporting summary**. Further information on research design is available in the Nature Research Reporting Summary linked to this article.

## Data availability
The data that support the findings of this study are available from the corresponding authors upon reasonable request. The source data for Figs. 4, 5 and Supplementary Figs. 4, 6–8, 10–13, and 16 and 17 are provided as a Source Data file.

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

## Acknowledgements

Funding was provided by the National Institutes of Health grant numbers P01-DK011794 (T.J.G.), P30-AR066261 (T.J.G.), 1DP1AI150593 (H.L.P.) and R01-AI087879 (H.L.P.). R.W.C. is supported in part by funding from the Cancer Research Institute Irvington Postdoctoral Fellowship. A.W.W. is supported by the Arnold O. Beckman Postdoctoral Fellowship. We acknowledge Jean-Pierre Vilardaga at the University of Pittsburgh Medical Center for generously providing the plasmid for β-Arr2-YFP. We acknowledge Ashok Khatri at Massahusetts General Hospital for providing PTH(1-20)-fluorescein and PTH(1-34)-TMR conjugates. We acknowledge Thomas Dean at Massachusetts General Hospital for technical assistance with cell bioassays.

## Author contributions

The paper was written by R.W.C. and H.P. Conceptual input was provided by R.W.C., F.A.F., A.W.W., T.G., and H.L.P. Experiments were designed and performed by R.W.C., F.A.F., E.D., N.S., and T.G.

## Competing interests

R.W.C., T.G., and H.L.P. have filed a patent (U.S. Provisional Application Serial No. 62/814,096) covering the use of V.H.H. fusions for targeting GPCRs. F.A.F., A.W.W., E.D, and N.S. declare no competing interests.
