## [Peer Review File · Nature Communications]

Reviewers' Comments:

Reviewer #1:

Remarks to the Author:

The manuscript by Cheloha et al describes the use of a new type of conjugate consisting of nanobodies linked to bioactive peptides to selectively target GPCRs. This is a novel and elegant study in which they show that the engineered nanobody-peptide conjugates show enhanced specificity and activity for PTHRs. The N-terminal peptide fragment of PTH conjugated to nanobodies targeting the PTHR show enhanced biological activity. While the natural peptides bind to both PTHR1 and PTHR2, these nanobodies-PTH peptide conjugates bind only to PTHR1. To couple nanobodies to peptides in C-C mode no genetic approach can be used and a sortase based approach was used to fuse the nanobodies to the N-terminal PTH peptides. Different engineered PTHRs (e.g insertion of GFP, YFP) and WT PTHR1 were used to show that they can bind nanobodies targeting the inserts within the engineered PTHR1 or ECD of the wildtype receptor.

- This approach is put forward as a new generic approach to generate ligands for cell surface receptors with improved properties. In this study the authors focus on the PTHR. To strengthen the findings in these studies to support the generic approach the addition of another example is recommended or suggested to indicate this could potentially lead to a CLAMP platform.

- In these studies the VHH-PTH peptide conjugates were tested in G protein dependent assays. For GPCRs, including PTHRs, however it is known that G protein independent signaling through barrestin plays an important role in biased signaling. Ligands binding to GPCRs may have biased properties. The VHH-PTH peptide conjugates show enhanced agonistic activity in G protein mediated signaling (cAMP, calcium mobilisation). What is the effect of the VHH-PTH peptides on barrestin recruitment in PTHR expressing cells?

- Signaling through the PTHR is known to be associated with sustained signaling, in part attributed to signaling from endosomal compartments. Slow-dissociating agonist PTH1-34 peptides were previously shown to form a very stable complex with the PTHR and consequently co-internalizes into endosomes persistently generating a cAMP response. Fast-dissociating agonist PTHR peptides for example do not co-internalize with the PTHR and only evokes cAMP production from the plasma membrane.

Since the VHH-PTH peptides described in this study show enhanced agonistic properties and delayed kinetics (see supporting figure on calcium mobilisation fig 11c vs 11a). Do you have evidence that signaling might occur from endosomal stores as well for the VHH-PTH peptide conjugates? One may consider using IF or use of biosensors targeting endosomes, or use of inhibitors of internalization.

- The authors generated VHH-peptide conjugates. Why not consider generating PTH peptide – VHH conjugates using a genetic approach? Bivalent or bispecific VHs, coupled by a flexible linker, bind two or more receptors.

Minor points:

- The constructs produced in these studies target PTHR1 and not PTHR2. Can you elaborate in the introduction/discussion whether for this receptor this would be beneficial.

- Fig 2a: explain in the main text what is meant M-PTH. We presume murine PTH?

- Fig 2b: should be twice VHH and not antibody

- Fig 3a: the colour of the tracing for the VHH-PTHR and VHH-6E should be reversed.
- Fig 3a: Please include quantitative axis on the FACS plot.
- include in IF staining fluorescence on cells that do not express PTHR1 and/or delta ECD expressing cells or PTHR2.
- Supporting figure 3: are the data included binding affinities? The curves should have a plateau to determine the actual K_d.
- We suggest to include the data presented in supplementary fig 5 d and e into figure 4 of the manuscript. These data are important to support the main findings.
- Shouldn't supporting Figure 6 be presented before suppl fig 5?
- Supporting fig 12. Explain in text why data on rat PTHR were added, while effects in a murine model are shown. The authors show that the VHH-peptide binds rat PTHR and thus presumably also to murine PTHR. Please include data showing that this VHH-peptide binds to murine PTHR or explain.
- Fig 4a x-axis in both graphs should be the same
- The title is very general. The part 'genetically impossible nanobody-peptide fusion' should not be inserted into the title.

Reviewer #2:

Remarks to the Author:

This elegant paper from the Ploegh lab provides a proof-of-principle for the novel use of a nanobody (VHH) to enhance the potency and specificity of a peptide ligand of a GPCR. The authors use PTHR1, a GPCR with a relatively large extracellular domain (ECD), truncated and modified peptides M-PTH11 or M-PTH14 derived from the natural ligand PTH(1-34) (used also as the drug teriparatide to treat osteoporosis), and the PTHR1-specific VHH 22A3 derived from a patent published by Ablynx. Using HEK cells stably transfected with PTHR1 and cAMP-activated luciferase, the authors show that the potency of M-PTH11 and M-PTH14 can be enhanced ~60-100 fold by chemical C-to-C conjugation to VHH 22A3 (VHHPTHR1). The potency of the VHH-PTH14 conjugate (EC₅₀ 0.075 nM) on these cells even surpasses that of the natural ligand PTH(aa1-34) (EC₅₀ 0.5 nM). Moreover, conjugation to the VHH dramatically enhances the specificity of PTH14 for PTHR1 over PTHR2 (inactive at 330 nM). Finally, the authors present the results of an in vivo experiment, in which they injected mice subcutaneously with peptide or peptide-VHH conjugates at 35 nmol/kg, followed by hourly measurements of ionized calcium levels in blood. The results indicate that the VHH-PTH14 conjugate may have stronger potency than PTH14 also in vivo, albeit only at one of five time points analyzed (1.36 mM vs. 1.29 mM ionized Ca, 2h after injection). In this experiment, the potency of the VHH-peptide conjugates approaches but does not surpass that of PTH(1-34). The authors present supporting complementary data using 6Etag- and GFP-specific VHHs (VHH-6E/VHH6E and VHH-enhancer/VHHGFP) and HEK cells stably transfected with recombinant variants of PTHR1 carrying genetic insertions of the 6E-tag or GFP protein in an exposed surface loop. Conjugation of the M-PTH11 or M-PTH14 peptides to these VHHs increased their potencies at the respective recombinant PTHR1 up to 7.800 fold.

With the exception of the in vivo experiments the results are solid and well explained. The paper could benefit from a discussion of certain limitations and more modest phrasing of some conclusions as detailed below.

Major concerns:

1. PTHR1 is a GPCR with a relatively large extracellular domain (ECD). In order for the technology to work, the VHH must not interfere sterically with binding of the peptide ligand. Most of the published chemokine-receptor-specific VHHs prevent binding of the natural peptide ligand (CXCR2, CXCR4, CXCR7). Many other GPCRs have smaller ECDs than PTHR1. This limitation should be discussed. Alternatively, additional supporting evidence could be presented for a GPCR with a smaller ECD.

2. Have the authors tested other PTHR1-specific VHHs? The Ablynx patent (reference 23) provides the sequences of 23 PTHR1-specific VHHs (representing 14 different VHH families). Have the authors examined other PTHR1-specific VHHs? How many of these enhanced/did not enhance the potency of PTH-peptides? If they did not examine other PTHR1-specific VHHs, please justify the choice of VHH22-A3.

3. The data on recombinant PTHR1 carrying a genetic insertion of the 6E-peptide or of GFP are solid and reinforce the data obtained with PTHR1. However, these insertions provide an "artificial landing site" (6E, GFP) and increase the size of the already large ECD of PTHR1 (GFP) and thereby the "landing space" for the VHH. The data on these insertions thus do not really provide any important additional insights. Occasionally, extensive presentation of data on VHH6E and VHHGFP actually distracts from the more important data on VHHPRTH1 (e.g. Fig. 1 and Table 1). The clarity of the paper could be enhanced by moving most of the data on VHH6E and VHHGFP to the Supporting Figures.

4. The conclusions of the study would be stronger if the authors could provide corresponding results for a second GPCR with a smaller ECD. Have the authors examined other GPCRs? The same Ablynx patent (reference 23) also provides the sequences of other peptide-activated GPCRs with smaller extracellular domains (CXCR4, MC4R).

5. The title contains the phrase "genetically impossible nanobody-peptide fusions". Granted, it is impossible to genetically fuse peptides carrying artificial amino acids to a nanobody. In the introduction, the authors state that "The site of antigen recognition on VHHs is near the N-terminus and the interaction of PTHR1 and PTHR2 with their ligands requires a free N-terminus on the latter." As expected, N-to-C VHH-peptide fusions did not have any detectable activity. VHHs, however, generally do tolerate N-terminal fusions without compromising antigen recognition. The literature abounds with examples of nanobody dimers and trimers fused by shorter and longer G4S linkers in which each nanobody module retains antigen recognition. The natural genetic fusions to examine would have been N-to-C fusions of PTH peptides to PTHR1-specific VHH, via a flexible linker of 5-15 aas. The authors should at least discuss this possibility. "Genetically impossible" should be omitted from the title.

6. Compared to the sound and extensive presentation of in vitro data, the results of the in vivo experiments are sparse. Fig. 5 shows the results of a single experiment with four groups of mice (n = 4 per group) which received subcutaneous injections of vehicle, peptide, or peptide-VHH conjugates at 35 nmol/kg, followed by hourly measurements of ionized calcium levels in blood. The Y-axis indicating the concentration of ionized Ca ranges in scale from 1.2-1.45 mM, exaggerating the effects. The cut-off on the Y-axis should be indicated in the legend and in the schematic, e.g. by insertion of a double line break in the Y axis. The asterisks in the schematic indicate significant differences vs. vehicle. The crucial p value for peptide(1-14) vs. VHH-peptide(1-14) conjugate should be indicated, preferably with the calculated value.

The curves for PTH-14 and the VHH-PTH14 conjugate overlap at four of five measured time points. The conjugate shows a higher value only at the 2 h time point. In the additional review material the authors state that the data were confirmed using independent experiments. The number of

independent experiments should be stated in the legend. Moreover, the paper would be strengthened by presenting the results of additional experiments in additional panels to Fig. 5, e.g. using other doses of compounds, additional time points, and a comparison of M-PTH(1-11) vs. VHHPTHR1(1-11).

7. Disulfide linkages in antibody-drug conjugates are known to be unstable in vivo. Does this pertain also to the C-to-C fusions of VHH and peptides used in this study?

8. The 5th sentence of the abstract should be softened. In "These C-to-C conjugates show biological activity vastly superior to that of the parent peptide, both in vitro and in vivo." please replace "vastly superior" with "~60-100 fold more potent". Also replace ", both in vitro and in vivo" by "in vitro". Then briefly summarize the in vivo findings in a separate sentence.

Minor concerns:

1. In the second sentence of the abstract "G protein-coupled receptors (GPCRs)" should be replaced by "a G protein-coupled receptor (GPCR)".

2. Fig. 3a: The color coding (blue and purple) of VHHPHR1 and VHH6E is reversed in the inset.

3. Fig. 5: Does "M-PTH(1-14)" differ from "PTH(1-14)" used in previous figures? If not, please harmonize the nomenclature. If so, please explain the difference.

4. p11: The § presenting the data of table I is cumbersome to read. It includes a flurry of values for the less important constructs (i.e. 1,800-7,800 fold more potency for conjugates acting on recombinant PTHR1), but omits the corresponding values for the important constructs i.e. conjugates acting on wildtype PTHR1. The § should focus on the results with VHHPTHR-conjugates vs. PTH(1-11) and PTH(1-14) peptides (57-fold and 103-fold more potent). Most of the data on PTHR16E and PTHR1GFP could be moved to the Supporting Figures. A brief summary without detailed values would suffice in the text.

5. p. 12, bottom of 1st §: The following sentence is difficult to understand and should be rephrased: "The enhancements in signaling activity provided by VHH conjugation is not seen with PTHR1 ligands that retain their ECD binding element: conjugates of PTH(1-34) and VHHs retain potent biological activity regardless of whether the target of the VHH is present on the cell line tested, at least at receptors with the ECD present (Supporting Figure 10)."

6. The 2nd and 3rd sentence of the last § of the discussion should be rephrased: "The ability to independently modulate receptor affinity. The structure of the agonist used to activate signaling should enable a further dissection of connections between ligand affinity, receptor signaling kinetics, and ligand bias."

7. Methods section. The recommended daily dosage of teriparatide/PTH(1-34) for humans is 20µg/injection. Please specify the dosage used (35 nmol/kg) in the mouse experiments as µg/injection. Justify the choice of the chosen dosage.

8. Please show an alignment VHH-22A3, VHH-05, and VHH-GFP as a Supporting Figure.

9. Supp. Fig. 5: Please replace "as described in methods" with a brief summary of the experimental procedure.

10. Supp. Figs. 5, 7, and 9: The use of different colors and symbols is confusing. Please make the figure easier to understand by pairing colors and symbols, e.g. use the same symbol for all peptides and a different symbol for all VHH-peptide conjugates. Use the same color for corresponding pairs of peptides and VHH-peptide conjugates.

11. Supp. Fig. 7 b) "removal of medium containing from ligand (right) in hPRTHR1 expressing HEK293 cells"

do you mean "removal of medium containing ligand (right) from hPRTHR1-expressing HEK293 cells"?

12. Supp. Fig. 13: Please replace the sequence "Align" by the sequence of PTHR1GFP (indicating only the first 5 and last 5 amino acids of the GFP sequence). Color-code the amino acids of the transmembrane domains and explain in the legend that the sequences have been truncated at position 240 after the second transmembrane domain. Move the cyan highlighting and underlining of exon2 to the sequence of hPTH1R (the 6E tag is not encoded by exon 2 and is not part of the crystal structure).

Responses to reviewer comments are in red

Reviewer #1 (Remarks to the Author):

The manuscript by Cheloha et al describes the use of a new type of conjugate consisting of nanobodies linked to bioactive peptides to selectively target GPCRs. This is a novel and elegant study in which they show that the engineered nanobody-peptide conjugates show enhanced specificity and activity for PTHRs. The N-terminal peptide fragment of PTH conjugated to nanobodies targeting the PTHR show enhanced biological activity. While the natural peptides bind to both PTHR1 and PTHR2, these nanobodies-PTH peptide conjugates bind only to PTHR1. To couple nanobodies to peptides in C-C mode no genetic approach can be used and a sortase based approach was used to fuse the nanobodies to the N-terminal PTH peptides. Different engineered PTHRs (e.g insertion of GFP, YFP) and WT PTHR1 were used to show that they can bind nanobodies targeting the inserts within the engineered PTHR1 or ECD of the wildtype receptor.

- This approach is put forward as a new generic approach to generate ligands for cell surface receptors with improved properties. In this study the authors focus on the PTHR. To strengthen the findings in these studies to support the generic approach the addition of another example is recommended or suggested to indicate this could potentially lead to a CLAMP platform.

We agree that the CLAMP platform deserves to be explored for other receptor systems but this is an area of study for future work. One major difference between PTHR1 and many other GPCRs is the size of the extracellular domain (ECD). B-family GPCRs possess ECDs larger than many members of A-family GPCRs (see comment by reviewer 2). To explore whether the CLAMP platform might be applicable to family A GPCRs, we have included new experiments. We show that a variant of PTHR1 lacking its entire ECD and carrying an exposed a peptide epitope tag instead, can be targeted using the CLAMP approach (Supporting Figure 12). This finding suggests that the CLAMP platform will be widely applicable, to be addressed in future studies.

- In these studies the VHH-PTH peptide conjugates were tested in G protein dependent assays. For GPCRs, including PTHRs, however it is know that G protein independent signaling through barrestin plays an important role in biased signaling. Ligands binding to GPCRs may have biased properties. The VHH-PTH peptide conjugates show enhanced agonistic activity in G protein mediated signaling (cAMP, calcium mobilisation). What is the effect of the VHH-PTH peptides on barrestin recruitment in PTHR expressing cells?

Selectivity for the induction of signaling through the various GPCR-mediated pathways (ligand bias) is indeed of interest. We have added new data showing that a VHH-PTH conjugate stimulates the recruitment of beta-arrestin, with behavior indistinguishable from PTH(1-34) (Supporting Figure 14). We have also added new text in the discussion on ligand bias.

- Signaling through the PTHR is known to be associated with sustained signaling, in part attributed to signaling from endosomal compartments. Slow-dissociating agonist PTH1-34 peptides were previously shown to form a very stable complex with the PTHR and consequently co-internalizes into endosomes persistently generating a cAMP response. Fast-dissociating agonist PTHR peptides for example do not

co-internalize with the PTHR and only evokes cAMP production from the plasma membrane.

Since the VHH-PTH peptides described in this study show enhanced agonistic properties and delayed kinetics (see supporting figure on calcium mobilisation fig 11c vs 11a). Do you have evidence that signaling might occur from endosomal stores as well for the VHH-PTH peptide conjugates? One may consider using IF or use of biosensors targeting endosomes, or use of inhibitors of internalization.

We have added new data to show that a VHH-PTH conjugate induces internalization (Supporting Figure 14 and 15). Using a PTHR1-GFP fusion with a pH-sensitive GFP, we show that VHH_{PTHR}-PTH(1-14) indeed changes the spectral characteristics of receptor-fused GFP (Figure 15) consistent with endosomal delivery. In this assay VHH_{PTHR}-PTH(1-14) behaves similarly to PTH(1-34), an agonist known to induce PTHR1 internalization. We furthermore visualized changes in the distribution of β -arrestin upon provision of ligand (Supporting Figure 14). β -arrestin recruitment is known to function in receptor internalization and results in colocalization with internalized receptor-ligand complexes. In this experiment VHH_{PTHR}-PTH(1-14) also behaves similarly to PTH(1-34). In VHH_{PTHR}-PTH(1-14) treated cells punctate signals from β -arrestin are observed near the nucleus, further supporting the conclusion that VHH_{PTHR}-PTH(1-14) induces internalization.

Testing whether these internalized ligand-receptor complexes are signaling-competent is not straightforward. Past work has shown a correlation between long acting peptide agonists and signaling-competent internalized ligand-receptor complexes. However, there is no unambiguous method to show that these internalized complexes remain signaling-competent, at least for PTHR1. Endosomal signaling has been inferred from signaling that persists after most receptor ligand complexes have moved from the cell surface into endosomal compartments. We show that PTH-VHH conjugates induce cAMP responses that persist for prolonged periods, which we attribute to signaling-competent internalized receptor-ligand complexes (Supporting Figure 8). We cannot use the nanobody biosensors described by Van Zastrow et al., because these are specific for other GPCRs and do not bind activated PTHR1. Other BRET-based biosensors based on colocalization of activated receptor would require modification of the intracellular portion of PTHR1 or of the G-protein with a reporter protein that could alter trafficking properties. Standard inhibitors of endocytosis (Dynasore, Pitstop) fail to alter the trafficking of PTHR1 following activation (data not shown).

The authors generated VHH-peptide conjugates. Why not consider generating PTH peptide – VHH conjugates using a genetic approach? Bivalent or bispecific VHs, coupled by a flexible linker, bind two or more receptors.

We did not pursue genetic VHH-peptide conjugates for two main reasons: i) we wished to retain maximum flexibility for the synthesis of many analogues from individual VHH constructs and ii) the need for free N-termini on PTH and possibly VHs. We conjugated several different synthetic PTH fragments to each VHH using chemoenzymatic methods. A genetic approach would require expression and purification of different constructs for each conjugate of interest. Such an effort would require extensive optimization, a substantial dedication of time and resources, and would not allow easy incorporation of residues such as aminobutyric acid (see Figure 2) .

Regarding the need for free N-termini, we showed that conjugation of a PTH fragments to the C terminus of a VHH eliminated PTHR1 agonist activity entirely, underscoring the need for a free N terminus for PTH (Supporting Figure 7) and its fragments. VHs can retain activity when fused to

another protein via the VHH N terminus. However, the N terminus is located near the site of antigen binding and we sought to avoid any possible disruption of antigen binding through modification of the VHH N terminus.

We address these points in the discussion section as follows “It is possible that the genetic fusion of PTH peptides to the N-terminus of VHHs might be accommodated with retention of both VHH binding and PTH activity. This would require a unique genetic construct and optimization of expression for each fusion and be incompatible with the introduction of non-natural amino acids such as aminobutyric acid. We avoid these drawbacks through our chemoenzymatic approach.”

Minor points:

- The constructs produced in these studies target PTHR1 and not PTHR2. Can you elaborate in the introduction/discussion whether for this receptor this would be beneficial.

We have added text to the discussion to expand on this topic. “PTHR1 is known to mediate the biological activity of PTH in treating osteoporosis, whereas the function of PTHR2 is more obscure. Tools to selectively target PTHR1, and subtypes of GPCRs in other families, will be useful for dissecting the biological function of receptors for which potent and selective ligands are of limited availability.”

- Fig 2a: explain in the main text what is meant M-PTH. We presume murine PTH?

M-PTH refers to a modified PTH analogue previously described. This is now clarified in the figure caption.

- Fig 2b: should be twice VHH and not antibody

This mistake has been corrected.

- Fig 3a: the colour of the tracing for the VHH-PTHR and VHH-6E should be reversed.

This mistake has been corrected.

- Fig 3a: Please include quantitative axis on the FACS plot.

This has been inserted.

- include in IF staining fluorescence on cells that do not express PTHR1 and/or delta ECD expressing cells or PTHR2.

This data has been added in Supporting Figure 5b.

- Supporting figure 3: are the data included binding affinities? The curves should have a plateau to determine the actual Kd.

We describe these data as half-maximal staining values and not Kds. We modified the methods section to describe how we estimated the maximal staining levels “For curves that did not reach plateau at the highest concentrations tested, curves were constrained by setting the maximal plateau value equal to that seen when staining that cell line with other VHHs that did achieve a plateau.”

- We suggest to include the data presented in supplementary fig 5 d and e into figure 4 of the manuscript. These data are important to support the main findings.

We appreciate this suggestion. We agree that the data in former Supplementary figure 5d-e (Now supplementary figures 6d-e) are important, but in order to minimize the amount of different data sets shown in Figure 4 we propose to leave this Supplementary data separate. We believe including only a few analogues in Figure 4 will make it easier to interpret and it more clearly demonstrates the benefit in selectivity shown by VHH_{PTHR} conjugation. We note that Reviewer 2 (comment 3) requests that we reduce the amount of data shown in the main figures of the manuscript.

- Shouldn't supporting Figure 6 be presented before suppl fig 5?

In the current version of the text we refer to Supporting Figure 6 (formerly Supporting Figure 5, just below Figure 3) before Supporting Figure 7 (formerly Supporting Figure 6, following page). It's unclear if there's another reason that the order of the figures should be reversed.

- Supporting fig 12. Explain in text why data on rat PTHR were added, while effects in a murine model are shown. The authors show that the VHH-peptide binds rat PTHR and thus presumably also to murine PTHR. Please include data showing that this VHH-peptide binds to murine PTHR or explain.

Just after Figure 4 we state "Since VHH_{PTHR} bound rat PTHR1 (Figure 3) we were confident that it would also bind mouse PTHR1, as these receptors are nearly identical (99% identical in their extracellular domain)."

- Fig 4a x-as in both graphs should be the same

The X-axes differ because of differences in the activity of peptides at these receptors. This has been clarified in the legend "Note that the x-axes in these graphs differ as peptides exhibit weaker activity for PTHR2."

- The title is very general. The part 'genetically impossible nanobody-peptide fusion' should not be inserted into the title.

We have altered the title to read "Improved GPCR ligands using nanobody tethering."

Reviewer #2 (Remarks to the Author):

This elegant paper from the Ploegh lab provides a proof-of-principle for the novel use of a nanobody (VHH) to enhance the potency and specificity of a peptide ligand of a GPCR. The authors use PTHR1, a GPCR with a relatively large extracellular domain (ECD), truncated and modified peptides M-PTH11 or M-PTH14 derived from the natural ligand PTH(1-34) (used also as the drug teriparatide to treat osteoporosis), and the PTHR1-specific VHH 22A3 derived from a patent published by Ablynx. Using HEK cells stably transfected with PTHR1 and cAMP-activated luciferase, the authors show that the potency of M-PTH11 and M-PTH14 can be enhanced ~60-100 fold by chemical C-to-C conjugation to VHH 22A3 (VHHPTHR1). The potency of the VHH-PTH14 conjugate (EC50 0.075 nM) on these cells even surpasses

that of the natural ligand PTH(aa1-34) (EC50 0.5 nM). Moreover, conjugation to the VHH dramatically enhances the specificity of PTH14 for PTHR1 over PTHR2 (inactive at 330 nM). Finally, the authors present the results of an in vivo experiment, in which they injected mice subcutaneously with peptide or peptide-VHH conjugates at 35 nmol/kg, followed by hourly measurements of ionized calcium levels in blood. The results indicate that the VHH-PTH14 conjugate may have stronger potency than PTH14 also in vivo, albeit only at one of five time points analyzed (1.36 mM vs. 1.29 mM ionized Ca, 2h after injection). In this experiment, the potency of the VHH-peptide conjugates approaches but does not surpass that of PTH(1-34). The authors present supporting complementary data using 6Etag- and GFP-specific VHHs (VHH-6E/VHH6E and VHH-enhancer/VHHGFP) and HEK cells stably transfected with recombinant variants of PTHR1 carrying genetic insertions of the 6E-tag or GFP protein in an exposed surface loop. Conjugation of the M-PTH11 or M-PTH14 peptides to these VHHs increased their potencies at the respective recombinant PTHR1 up to 7.800 fold.

With the exception of the in vivo experiments the results are solid and well explained. The paper could benefit from a discussion of certain limitations and more modest phrasing of some conclusions as detailed below.

Major concerns:

1. PTHR1 is a GPCR with a relatively large extracellular domain (ECD). In order for the technology to work, the VHH must not interfere sterically with binding of the peptide ligand. Most of the published chemokine-receptor-specific VHHs prevent binding of the natural peptide ligand (CXCR2, CXCR4, CXCR7). Many other GPCRs have smaller ECDs than PTHR1. This limitation should be discussed. Alternatively, additional supporting evidence could be presented for a GPCR with a smaller ECD.

We endorse the reviewer's suggestion that there are likely differences between GPCRs with varying sizes of ECDs and mechanisms of interaction with ligand. We have added new data which shows that replacement of the large ECD of PTHR1 with a small epitope tag (6E tag) enables activation of this receptor by use of the VHH that binds to the 6E tag (Supporting Figure 12). The fact that this approach is effective in a receptor construct with a small ECD suggests that the CLAMP platform could be applied to receptors with smaller ECDs.

2. Have the authors tested other PTHR1-specific VHHs? The Ablynx patent (reference 23) provides the sequences of 23 PTHR1-specific VHHs (representing 14 different VHH families). Have the authors examined other PTHR1-specific VHHs? How many of these enhanced/did not enhance the potency of PTH-peptides? If they did not examine other PTHR1-specific VHHs, please justify the choice of VHH22-A3.

We have not tested other VHHs from the Ablynx patent. We chose VHH22-A3 as data from this reference suggested this antibody had the highest affinity for PTHR1 of any VHH identified. This is now mentioned in the methods section. It will be of interest for future studies to test whether nanobodies that bind to different epitopes on the same receptor affect the ability of such VHH-peptide conjugates to activate receptors.

3. The data on recombinant PTHR1 carrying a genetic insertion of the 6E-peptide or of GFP are solid and

reinforce the data obtained with PTHR1. However, these insertions provide an "artificial landing site" (6E, GFP) and increase the size of the already large ECD of PTHR1 (GFP) and thereby the "landing space" for the VHH. The data on these insertions thus do not really provide any important additional insights. Occasionally, extensive presentation of data on VHH6E and VHHGFP actually distracts from the more important data on VHHPRTH1 (e.g. Fig. 1 and Table 1). The clarity of the paper could be enhanced by moving most of the data on VHH6E and VHHGFP to the Supporting Figures.

We appreciate this perspective and we have moved data corresponding to PTHR1-GFP to the supporting information to reduce distraction. We have left data for other constructs in the main text as we believe this provides an important proof of concept and will be instructive for the future design of designer receptors specific for artificial ligands.

4. The conclusions of the study would be stronger if the authors could provide corresponding results for a second GPCR with a smaller ECD. Have the authors examined other GPCRs? The same Ablynx patent (reference 23) also provides the sequences of other peptide-activated GPCRs with smaller extracellular domains (CXCR4, MC4R).

This is clearly an area for future research. We have added data showing that a receptor with a minimal ECD can be targeted using this approach (Supporting Figure 12), further supporting the suggestion that this approach will be broadly applicable.

5. The title contains the phrase "genetically impossible nanobody-peptide fusions". Granted, it is impossible to genetically fuse peptides carrying artificial amino acids to a nanobody. In the introduction, the authors state that "The site of antigen recognition on VHHs is near the N-terminus and the interaction of PTHR1 and PTHR2 with their ligands requires a free N-terminus on the latter." As expected, N-to-C VHH-peptide fusions did not have any detectable activity. VHHs, however, generally do tolerate N-terminal fusions without compromising antigen recognition. The literature abounds with examples of nanobody dimers and trimers fused by shorter and longer G4S linkers in which each nanobody module retains antigen recognition. The natural genetic fusions to examine would have been N-to-C fusions of PTH peptides to PTHR1-specific VHH, via a flexible linker of 5-15 aas. The authors should at least discuss this possibility. "Genetically impossible" should be omitted from the title.

The title has been changed to incorporate this suggestion. We have added text to the discussion section to highlight the point raised in this comment "It is possible that the genetic fusion of PTH peptides to the N-terminus of VHHs might be accommodated with retention of both VHH binding and PTH activity. This would require a unique genetic construct and optimization of expression for each fusion and be incompatible with the introduction of non-natural amino acids such as aminobutyric acid. We avoid these drawbacks through our chemoenzymatic approach."

6. Compared to the sound and extensive presentation of in vitro data, the results of the in vivo experiments are sparse. Fig. 5 shows the results of a single experiment with four groups of mice (n = 4 per group) which received subcutaneous injections of vehicle, peptide, or peptide-VHH conjugates at 35 nmol/kg, followed by hourly measurements of ionized calcium levels in blood. The Y-axis indicating the concentration of ionized Ca ranges in scale from 1.2-1.45 mM, exaggerating the effects. The cut - off on

the Y-axis should be indicated in the legend and in the schematic, e.g. by insertion of a double line break in the Y axis. The asterisks in the schematic indicate significant differences vs. vehicle. The crucial p value for peptide(1-14) vs. VHH-peptide(1-14) conjugate should be indicated, preferably with the calculated value.

We have now added a double line break in the figure to indicate the discontinuity in the Y-axis, although we note that this style of data depiction is in line with common practices for this assay. Depiction of a narrow range is needed to illustrate the effects of PTH ligand injection on blood Ca^{2+} , which are typically ~10% or less starting from baseline (1.2 mM). We have also added text in the legend to highlight this line break. We have now used distinct symbols to indicate exact p values for each comparison.

The curves for PTH-14 and the VHH-PTH14 conjugate overlap at four of five measured time points. The conjugate shows a higher value only at the 2 h time point. In the additional review material the authors state that the data were confirmed using independent experiments. The number of independent experiments should be stated in the legend. Moreover, the paper would be strengthened by presenting the results of additional experiments in additional panels to Fig. 5, e.g. using other doses of compounds, additional time points, and a comparison of M-PTH(1-11) vs. VHHPTH1(1-11).

We only refer to independent experiments in individual figure captions where independent replication experiments were performed. The experimental setup in Figure 5 was not replicated in multiple experiments, as the data generated were statistically significant (see revised figure 5 for expanded details about statistical significance). Therefore, we don't believe a comment on independent experiments is needed for this caption. We used mouse cohort sizes based on past experience in both our laboratory and outside for this particular experiment. While it would be interesting to test other doses and compounds, they would not alter the main conclusion from this experiment: a VHH-peptide conjugate shows biological activity *in vivo* whereas the free peptide does not. We therefore suggest that further experiments are not needed for this manuscript.

7. Disulfide linkages in antibody-drug conjugates are known to be unstable *in vivo*. Does this pertain also to the C-to-C fusions of VHH and peptides used in this study?

Disulfide linkages are susceptible to cleavage through reduction by thiols and other reducing agents. The C-to-C fusions used in this study are linked by triazole linkages, which are known to be stable under physiological conditions. We have added a sentence to highlight this point "Of note, the resulting triazole linkage is not susceptible to cleavage by reduction, unlike the disulfide linkages used in other conjugates."

8. The 5th sentence of the abstract should be softened. In "These C-to-C conjugates show biological activity vastly superior to that of the parent peptide, both *in vitro* and *in vivo*." please replace "vastly superior" with "~60-100 fold more potent". Also replace ", both *in vitro* and *in vivo*" by "*in vitro*". Then briefly summarize the *in vivo* findings in a separate sentence.

We removed the word "vastly" from this sentence to avoid oversimplification. We note that in some cases VHH conjugation enhances activity by >1800 fold (compare PTH(1-10) to VHH_{6E}-PTH(1-10) on PTHR1 6E). We have clarified this statement to indicate that enhanced activity *in vivo* refers to a comparison between the peptide fragment and that same fragment conjugated to VHH. We note here

that the bioactivity of a PTH(1-14) conjugate is remarkable because it is the first time that we have observed such activity for a PTH peptide as short as PTH(1-14).

Minor concerns:

1. In the second sentence of the abstract "G protein-coupled receptors (GPCRs)" should be replaced by "a G protein-coupled receptor (GPCR)".

We implemented this change.

2. Fig. 3a: The color coding (blue and purple) of VHH_{PHR1} and VHH_{6E} is reversed in the inset.

We corrected this error.

3. Fig. 5: Does "M-PTH(1-14)" differ from "PTH(1-14)" used in previous figures? If not, please harmonize the nomenclature. If so, please explain the difference.

The sequence of M-PTH(1-14) is slightly different and it is now listed in the figure caption.

4. p11: The § presenting the data of table I is cumbersome to read. It includes a flurry of values for the less important constructs (i.e. 1,800-7,800 fold more potency for conjugates acting on recombinant PTHR1), but omits the corresponding values for the important constructs i.e. conjugates acting on wildtype PTHR1. The § should focus on the results with VHH_{PTHR}-conjugates vs. PTH(1-11) and PTH(1-14) peptides (57-fold and 103-fold more potent). Most of the data on PTHR16E and PTHR1GFP could be moved to the Supporting Figures. A brief summary without detailed values would suffice in the text.

We have removed most of the text discussing -fold enhancement enabled by VHH conjugations, highlighting only the most dramatic examples. We have also included a reference to the comparison between PTH(1-14) and VHH_{PTHR}-PTH(1-14) on WT PTHR1.

5. p. 12, bottom of 1st §: The following sentence is difficult to understand and should be rephrased: "The enhancements in signaling activity provided by VHH conjugation is not seen with PTHR1 ligands that retain their ECD binding element: conjugates of PTH(1-34) and VHHs retain potent biological activity regardless of whether the target of the VHH is present on the cell line tested, at least at receptors with the ECD present (Supporting Figure 10)."

We rephrased this passage as follows "The enhanced signaling activity provided by VHH conjugation is not seen with PTHR1 ligands that bind through both ECD and transmembrane domain interactions: conjugation of PTH(1-34) with VHHs yields active compounds, regardless of whether the target of the VHH is present on the cell line tested (Supporting Figure 11). VHH_{PTHR}-PTH(1-14) activated PTHR1YFPΔECD, even though the VHH does not bind to this receptor (Table 1).

6. The 2nd and 3rd sentence of the last § of the discussion should be rephrased:

"The ability to independently modulate receptor affinity. The structure of the agonist used to activate signaling should enable a further dissection of connections between ligand affinity, receptor signaling kinetics, and ligand bias."

This was a typographical error. We have corrected this passage as follows “The ability to modulate receptor affinity while not modifying the structure of the agonist used to activate signaling should enable a further dissection of connections between ligand affinity, receptor signaling kinetics, and ligand bias⁴⁷.”

7. Methods section. The recommended daily dosage of teriparatide/PTH(1-34) for humans is 20µg/injection. Please specify the dosage used (35 nmol/kg) in the mouse experiments as µg/injection. Justify the choice of the chosen dosage.

We chose our dose based on past experience with this assay showing that this dose of PTH(1-34) would induce a calcemic response that would be manifest 1-2 hours after injection and dissipate by 6 hours after injection. This is now referenced in the methods section. Indicating dose scaled to body weight is common practice for PTH animal studies and so is used as the preferred method.

8. Please show an alignment VHH-22A3, VHH-05, and VHH-GFP as a Supporting Figure.

This has been added as Supporting Figure 18.

9. Supp. Fig. 5: Please replace "as described in methods" with a brief summary of the experimental procedure.

This figure caption has been updated.

10. Supp. Figs. 5, 7, and 9: The use of different colors and symbols is confusing. Please make the figure easier to understand by pairing colors and symbols, e.g. use the same symbol for all peptides and a different symbol for all VHH-peptide conjugates. Use the same color for corresponding pairs of peptides and VHH-peptide conjugates.

For each of these (now Supporting Figures 6, 8, and 10) we standardized closed and open squares for peptides and VHH-peptide conjugates, respectively. Matching colors was not possible as some peptides were conjugated to more than different VHH in a single panel.

11. Supp. Fig. 7 b) "removal of medium containing from ligand (right) in hPRTHR1 expressing HEK293 cells"

do you mean "removal of medium containing ligand (right) from hPRTHR1-expressing HEK293 cells"?

This has been corrected. Thank you for catching this error.

12. Supp. Fig. 13: Please replace the sequence "Align" by the sequence of PTHR1GFP (indicating only the first 5 and last 5 amino acids of the GFP sequence). Color-code the amino acids of the transmembrane domains and explain in the legend that the sequences have been truncated at position 240 after the second transmembrane domain. Move the cyan highlighting and underlining of exon2 to the sequence of hPTH1 (the 6E tag is not encoded by exon 2 and is not part of the crystal structure).

Since we have added new PTHR1 constructs we have consolidated all sequence data and annotation into a single figure (new Supporting Figure 17).

Reviewers' Comments:

Reviewer #1:

Remarks to the Author:

New data have been provided that demonstrate that the CLAMP approach can be used as a generic approach. The addition of an additional epitope tag on the PTHR1 receptor lacking the ECD domain, resulted in a receptor in which VHH6E-PTH1-11 showed increased potency on the PTHR1-delINT-6E receptor (supporting fig 12) compared to the PTH peptide. These data indicate that this approach can be used for other receptors too.

- Experiments showing that the VHH-PTH conjugates induce barrestin recruitment have been added (supporting Fig 14) and can induce internalisation (supporting Fig 15). The VHH-PTH conjugates induce barrestin recruitment. The expression of the receptors is only limited to a few cells. In a), b) and c) the authors show that the fluorescently labelled PTH peptide in cells expressing the receptor barrestin recruitment, as observed a punctuate staining, is observed. The data in d), e) and f), however, are inconclusive, as the data do not show that in the highlighted box cells are indeed expressing the receptor. As only a limited number of cells express the receptor one should show in d), e) and f) that those cells indeed express the receptors. Using labelled VHH-PTH conjugate (as done for a), b) c)) or antibodies targeting VHHs (anti-VHH) in cells expressing tagged PTHR1 receptor should resolve this.

- In supporting figure 4 (former supporting Fig 3) values are inserted in the 3 upper figures. Yet, as previously indicated by reviewer 2 the curves are not finished and therefore no values can be assigned. The authors have stated a plateau was estimated based on maximal labelling with other VHHs and represent staining values. Since labeling may vary among VHHs I suggest to remove the 3 upper values in these figures and refer in the text of legend for these values > 150 nM instead. This should be adjusted.

- The title has been changed to 'Improved GPCR ligands using nanobody tethering'. I suggest to include the word potencies to indicate their potencies are increased ... 'Improved potencies of GPCR ligands using nanobody tethering'.

Minor points:

- typo in legend of supporting Fig 14 HEK293- derved should be HEK293 derived

- Since new supporting figures have been added renumber the supplementary figures in the order they appear in the manuscript.

Reviewer #2:

Remarks to the Author:

In this revised version the authors address most of my concerns satisfactorily. I still have concerns regarding the second example of a GPCR and the in vivo experiment:

1. In lieu of a demonstration that their CLAMP technology works also on a second GPCR with a smaller extracellular domain, the authors provide new data using a new variant of PTHR1, in which the entire 161 aa N-terminal domain was replaced with the 14 aa E6 tag. The data provided in Supporting Figure 12 show that the VHH peptide(1-11) conjugate is ~50 fold more potent than the free peptide. This finding indicates that VHHs directed against a natural short N-terminal peptide of a GPCR could also serve as an anchoring point for the delivery of peptide ligands by the CLAMP technology.

Although similar data with a distinct GPCR certainly would strengthen the key message, I understand that this would entail a lot of extra work and look forward to future follow-up studies.

2. I am still struck by the disparity between the sparse in vivo data and the rich, high quality, in vitro data.

The authors now clarified that they performed only a single in vivo experiment and they now provide the p values for the results of this experiment in the legend of Figure 5: A single data point (at 2h) shows a significant difference between the VHH-peptide conjugate and the free peptide with a p-value of 0.038 at a sample size of n=4. The authors conclude that "a VHH-peptide conjugate shows biological activity in vivo whereas the free peptide does not" and hold that testing other doses, while interesting, would not alter this main conclusion.

This may be true. However, similar results obtained in a second independent experiment would verify that this key finding is reproducible and would thereby strengthen the conclusion. It should be noted that the dose of the VHH-peptide conjugate used in this experiment (35 nmol/kg) is rather high (600 µg/mouse or 30 mg/kg). By comparison, the therapeutic dose of the only VHH compound currently in the clinic, Caplacizumab, is much lower (10 mg/human = 0,17 mg/kg), even when converting mouse to human dose equivalents ($\times 12.3 = 2$ mg/kg) (see Nair et al 2016, PMID: 27057123).

I recommend that the authors perform a dose titration analyses of the VHH-peptide conjugate VHHPTHR-PTH(1-14) at the 2h time point with a larger sample size to strengthen their case.

Without such a second independent experiment, I consider the following amendments essential:

- clarify in the legend of Fig. 5 that only a single in vivo experiment was performed.
- in the reporting summary, p.2 life science study design, Replication: clarify that the following statement does not apply to the in vivo experiment:
Data were confirmed using independent experiments carried out the same laboratory as indicated in figure captions. These replication experiments were successful.
- in the abstract, distinguish the conclusion from the in vitro experiments and the single in vivo experiment, i.e. by separating the following sentence:
„These C-to-C conjugates show biological activity superior to that of the parent fragment peptide in vitro and in mice, as shown for a conjugate comprised of a PTH(1-14) analogue and the PTHR1-binding nanobody.“
to something like:
These C-to-C conjugates show biological activity superior to that of the parent fragment peptide in vitro. In an exploratory experiment in mice, a conjugate comprised of a PTH(1-14) analogue and the PTHR1-binding nanobody showed biological activity whereas the free peptide did not (p=0.038).

Minor points:

p.5 last sentence: please insert "N-terminal" behind "entire"
Another version of PTHR1 in which yellow fluorescent protein (YFP) replaces the entire extracellular domain (PTHR1YFPΔECD, Figure 1e)²¹, was also used.

p.12 third line, contains a duplication: „the the“

Reviewers' comments (responses in red)

Reviewer #1 (Remarks to the Author):

- Experiments showing that the VHH-PTH conjugates induce b-arrestin recruitment have been added (supporting Fig 14) and can induce internalisation (supporting Fig 15). The VHH-PTH conjugates induce b-arrestin recruitment. The expression of the receptors is only limited to a few cells. In a), b) and c) the authors show that the fluorescently labelled PTH peptide in cells expressing the receptor b-arrestin recruitment, as observed a punctuate staining, is observed. The data in d), e) and f), however, are inconclusive, as the data do not show that in the highlighted box cells are indeed expressing the receptor. As only a limited number of cells express the receptor one should show in d), e) and f) that those cells indeed express the receptors. Using labelled VHH-PTH conjugate (as done for a), b) c)) or antibodies targeting VHHs (anti-VHH) in cells expressing tagged PTHR1 receptor should resolve this.

We have now added new data in new Supporting Figure 15 in which HEK293 cells expressing β -arrestin2-YFP were transfected with a PTHR1-HA tag construct and treated with VHH_{PTHR}-PTH(1-14). Receptor was visualized with anti-HA-fluorophore after fixation and permeabilization. Transfected cells show punctate and colocalized β -arrestin2-YFP and PTHR1-HA signals but only upon VHH_{PTHR}-PTH(1-14) treatment. We conclude that VHH_{PTHR}-PTH(1-14) stimulates β -arrestin recruitment to PTHR1.

- In supporting figure 4 (former supporting Fig 3) values are inserted in the 3 upper figures. Yet, as previously indicated by reviewer 2 the curves are not finished and therefore no values can be assigned. The authors have stated a plateau was estimated based on maximal labelling with other VHHs and represent staining values. Since labeling may vary among VHHs I suggest to remove the 3 upper values in these figures and refer in the text of legend for these values > 150 nM instead. This should be adjusted.

This change has been implemented.

- The title has been changed to 'Improved GPCR ligands using nanobody tethering'. I suggest to include the word potencies to indicate their potencies are increased ... 'Improved potencies of GPCR ligands using nanobody tethering'.

We appreciate this comment. One important property of these conjugates is their improved **selectivity** relative to prototype peptides, which is not captured if only describing potencies. We therefore suggest that the title remain as written.

Minor points:

- typo in legend of supporting Fig 14 HEK293- derved should be HEK293 derived

This has now been corrected.

- Since new supporting figures have been added renumber the supplementary figures in the order they appear in the manuscript.

We have reviewed the ordering of references to supplementary figures in the main text and they now appear in the same order as the supplementary figures in the supporting information.

Reviewer #2 (Remarks to the Author):

It should be noted that the dose of the VHH-peptide conjugate used in this experiment (35 nmol/kg) is rather high (600 µg/mouse or 30 mg/kg). By comparison, the therapeutic dose of the only VHH compound currently in the clinic, Caplacizumab, is much lower (10 mg/human = 0,17 mg/kg), even when converting mouse to human dose equivalents ($\times 12.3 = 2$ mg/kg) (see Nair et al 2016, PMID: 27057123).

I recommend that the authors perform a dose titration analyses of the VHH-peptide conjugate VHHPTHR-PTH(1-14) at the 2h time point with a larger sample size to strengthen their case.

Without such a second independent experiment, I consider the following amendments essential:

- clarify in the legend of Fig. 5 that only a single in vivo experiment was performed.
- in the reporting summary, p.2 life science study design, Replication: clarify that the following statement does not apply to the in vivo experiment:
Data were confirmed using independent experiments carried out the same laboratory as indicated in figure captions. These replication experiments were successful.
- in the abstract, distinguish the conclusion from the in vitro experiments and the single in vivo experiment, i.e. by separating the following sentence:
„These C-to-C conjugates show biological activity superior to that of the parent fragment peptide in vitro and in mice, as shown for a conjugate comprised of a PTH(1-14) analogue and the PTHR1-binding nanobody.“
to something like:
These C-to-C conjugates show biological activity superior to that of the parent fragment peptide in vitro. In an exploratory experiment in mice, a conjugate comprised of a PTH(1-14) analogue and the PTHR1-binding nanobody showed biological activity whereas the free peptide did not ($p=0.038$).

We appreciate this point and the related reference. We note that we chose the dose for this experiment was based on precedent with peptide for stimulating calcemic responses in mice, and not for VHH-based therapies. This point has now been more clearly explained in the “In vivo activity” section of the text. We have also implemented the changes discussed after the outlined comment above to more clearly describe the *in vivo* experiment.

Minor points:

p.5 last sentence: please insert "N-terminal" behind "entire"
Another version of PTHR1 in which yellow fluorescent protein (YFP) replaces the entire extracellular domain (PTHR1YFPΔECD, Figure 1e)²¹, was also used.

This was inserted.

p.12 third line, contains a duplication: „the the“

This is now corrected.